# Metritocracy:
# Representative Metrics for Lite Benchmarks

**Ariel D. Procaccia**

Harvard University

**Benjamin Schiffer**

Harvard University

**Serena Wang**

Harvard University

**Shirley Zhang**

Harvard University

## Abstract

A common problem in LLM evaluation is how to choose a subset of metrics from a full suite of possible metrics. Subset selection is usually done for efficiency or interpretability reasons, and the goal is often to select a "representative" subset of metrics. However, "representative" is rarely clearly defined. In this work, we use ideas from social choice theory to formalize two notions of representation for the selection of a subset of evaluation metrics. We first introduce *positional representation*, which guarantees every alternative is sufficiently represented at every position cutoff. We then introduce *positional proportionality*, which guarantees no alternative is proportionally over- or under-represented by more than a small error at any position. We prove upper and lower bounds on the smallest number of metrics needed to guarantee either of these properties in the worst case. We also study a generalized form of each property that allows for additional input on groups of metrics that must be represented. Finally, we tie theory to practice through real-world case studies on both LLM evaluation and hospital quality evaluation.

## 1 Introduction

The last few years have seen an explosion in metrics to evaluate large language models (LLMs). While this has improved our ability to understand the capabilities of LLMs, it is also increasingly computationally expensive to evaluate all of these measures. This challenge is directly felt by platforms like BIG-bench [Srivastava et al., 2022] and HELM [Liang et al., 2022] that aggregate numerous metrics to provide as complete a picture as possible of LLM performance.

A common approach that evaluation platforms take to mitigate these growing computational difficulties is to create a "lite" version of the full evaluation suite, which consists of a subset of the original measures. For example, BIG-bench Lite contains a subset of 24 JSON metrics from the full collection of over 200 metrics, which is "designed to provide a canonical measure of model performance, while being far cheaper to evaluate than the full set." HELM Lite also contains a subset of scenarios from HELM Classic (in addition to some others), constructed to have a lighter computational overhead.

The problem of selecting a subset of evaluation metrics is actually quite general beyond LLM evaluation, and is also common in public policy and business operations. For example, Cal Hospital Compare awards Patient Safety Honor Roll status to hospitals using a subset of 12 measures from a full set of hundreds of hospital quality measures collected by the Centers for Medicare and Medicaid Services [Cal Hospital Compare, 2025]. This subset is carefully hand-selected, but Cal Hospital Compare still acknowledges that "measurement of patient safety is complex and there is no single validated method for measuring the overall safety of care provided in a given health care setting." Beyond computational considerations, an additional reason for selecting a subset of metrics is understandability for stakeholders.

39th Conference on Neural Information Processing Systems (NeurIPS 2025).

Across these settings, a recurring theme is that practitioners want to select a subset of metrics that is in some sense "representative" of the underlying full set of metrics.[1] However, there is no formal notion of representation that is common across these contexts. To provide the tools to more clearly discuss and achieve representation in the selection of a subset of evaluation metrics, we introduce formal definitions of representation inspired by notions in computational social choice. Our theoretical framework provides the basis for discussing tradeoffs between representation and computational cost or understandability, and allows practitioners to more clearly reason about the consequences of subset selection on downstream decision-making. Our framework also opens the door to algorithmic support for metric selection, which we characterize theoretically and empirically.

## 1.1 Our Contributions

We study the problem of selecting a representative subset from a set of $n$ evaluation metrics that each rank $m$ alternatives.[2] We introduce two desirable properties for the subset and provide lower and upper bounds on the number of metrics needed to satisfy each of the two properties.

We start by defining positional representation, which prevents under-representation. Positional representation guarantees that every alternative is sufficiently represented at every rank. This property is parameterized by a group size $g$, which indicates the granularity of representation. We show upper and lower bounds on the number of metrics necessary to satisfy positional representation in the worst case that are tight up to a logarithmic factor. We also provide a polynomial-time algorithm which always finds a subset satisfying positional representation with size at most $\frac{n}{g} \log(m)$.

We next introduce positional proportionality, which guarantees that no alternative is under- or over-represented at any position in the chosen subset by more than an additive factor of $\epsilon$. We give tight (up to constant factors) upper and lower bounds on the number of metrics needed to satisfy positional proportionality in the worst case. We also show that any subset of metrics satisfying positional proportionality can approximate any social choice scoring rule on the original set of metrics.

Finally, we generalize both properties to enable preserving information about the original set of metrics which may be external to the rank information. We show that our upper and lower bound extend to the general versions of our properties, and prove that finding the smallest set satisfying either general property is NP-hard. Our theoretical results are summarized in Table 1.

| Properties | Parameter | Upper Bound | Lower Bound | Complexity |
|---|---|---|---|---|
| **Positional Representation** | Group size $g$ | $O\left(\frac{n}{g} \log(m)\right)$ | $\Omega\left(\frac{\frac{n}{g} \log(m)}{\log(\frac{n}{g} \log(m))}\right)$ | NP-hard |
| **Positional Proportionality** | Accuracy $\epsilon$ | $O\left(\frac{1}{\epsilon^2} \log(m)\right)$ | $\Omega\left(\frac{1}{\epsilon^2} \log(m)\right)$ | NP-hard |

Table 1: Summary of theoretical results.

To connect these theoretical results to the above motivating practical examples, we evaluated algorithms to achieve positional representation and positional proportionality in three case studies with real data: two on LLM evaluation and one on hospital quality evaluation. We show that the outputs of our methods compare favorably with the existing subsets currently deployed in the real world for each of these settings.

## 1.2 Related Work

There has been a surge of recent work studying benchmarks as voters from the social choice perspective. These works differ from ours in that none study subset selection of metrics. Many of these works study how metrics should be aggregated, particularly to improve robustness [Colombo et al., 2022b,a, Peyrard et al., 2017, Mishra and Arunkumar, 2021, Himmi et al., 2023]. Colombo et al. [2022a] propose using Borda count to aggregate benchmarks as an approximation of the Kemeny Rule and Rofin et al. [2022] propose VOTE'N'RANK, which consists of several scoring rules based on social choice theory for benchmark aggregation. In a different direction, Zhang and Hardt [2024] give a

---

[1]Not all "Lite" benchmarks are trying to be representative — SWE-bench Lite [Jimenez et al., 2023] also tries to include easier metrics. In this work we focus on settings where the goal is to be representative.

[2]A *metric* refers to anything that gives a ranking over alternatives (and not necessarily a loss function).

form of Arrow's Impossibility result for the benchmark setting and highlight an inherent tradeoffs between sensitivity and diversity.

In social choice, our properties are most closely related to Justified Representation (JR) introduced by Aziz et al. [2017] in the committee selection setting. JR guarantees representation for sufficiently-large "coalitions" of voters that approve the same candidate; it is similar to our notion of positional representation, which guarantees representation for "coalitions" of benchmarks that all rank alternative $a$ in the top $r$. However, our setting has no voters and no notion of approval, which makes a direct comparison impossible. See Appendix C for more discussion of the relationship to JR. Also in the approval setting, Skowron and Faliszewski [2015] study the relationship between set cover and proportional representation with approval ballots, similar to our NP-hardness results in Section 4.

Other methods have also been proposed to speed up LLM evaluation through selection of individual prompts from all possible evaluation metrics [Perlitz et al., 2023, Polo et al., 2024, Li et al., 2024]. Our work differs in that we restrict to selecting a subset of full metrics, and not the more granular selection of prompts. This captures more general public policy settings like hospital quality evaluation, where only full metrics are available. Note, however, that the two approaches are complementary and can be used in tandem.

### 1.3 Model

Let there be a set $N = [n]$ of *metrics* and a set $A = [m]$ of *alternatives*. Each metric $i$ has a *ranking* $\sigma_i$ over alternatives. Let $\sigma_{ir}$ be the alternative ranked in position $r$ in metric $i$'s ranking, and let $\sigma_i(a)$ be the rank of alternative $a$ in metric $i$'s ranking. The set of all metric rankings forms a *preference profile* $\boldsymbol{\sigma}_N = \{\sigma_1, \ldots, \sigma_n\}$. For $K \subseteq N$, define $\boldsymbol{\sigma}_K = \{\sigma_i : i \in K\}$. We study the following:

> *How should we select a small subset of metrics $K \subset N$ such that $K$ preserves some information from the metrics in $N$?*

We say that metric $i$ ranks alternative $a$ at least at position $r$ if $\sigma_i(a)$ is $r$ or better (i.e. $\sigma_i(a) \leq r$). Similarly, we say that metric $i$ ranks alternative $a$ above position $r$ if $\sigma_i(a)$ is strictly better than $r$ (i.e. $\sigma_i(a) < r$). Define $C(N, r, a)$ as the number of metrics in $N$ that rank alternative $a$ in the top $r$. Likewise, for $K \subseteq N$, $C(K, r, a)$ is the number of metrics in $K$ that rank alternative $a$ in the top $r$.

## 2 Positional Representation

Consider some alternative $a \in A$. If $a$ is ranked highly by many metrics in $N$, then we want $a$ to be ranked highly in many metrics in the subset $K$ as well—otherwise, $a$ would not be getting the representation it deserves in $K$. Intuitively, it would be undesirable if $a$ is ranked in the top 10 positions by the majority of the metrics in $N$, but does not appear in the top 10 positions for any metric in $K$. Similarly, it would be undesirable if $b$ is ranked in the top 50 by 90% of metrics in $N$, but $b$ is ranked in the top 50 by less than half of the metrics in $K$. We therefore begin by introducing positional representation, which guarantees that the subset $K$ gives every alternative $a \in A$ sufficient representation at *every* cutoff position. More specifically, positional representation guarantees that for every position cutoff, if $a$ is ranked above the cutoff in a sufficiently large number of the original metrics, then that alternative is also ranked above the cutoff in a (close to) proportional number of metrics in $K$.

**Definition 1.** *A subset $K$ satisfies* positional representation *for group size $g$ if for every $r \in [1 : m]$, any alternative that is ranked in the first $r$ positions in at least $\ell \cdot g$ metrics is ranked in the first $r$ positions in at least $\ell$ metrics in $K$. Equivalently, for all $r \in [1 : m]$ and all $a \in A$,*

$$C(K, r, a) \geq \left\lfloor \frac{C(N, r, a)}{g} \right\rfloor. \tag{1}$$

Positional representation is parameterized by a group size $g$, which will capture the tradeoff between the granularity of representation and the size of $K$ needed to satisfy positional representation. As $g$ gets larger, the minimum necessary $|K|$ decreases, but for each alternative $a$ it takes more high-ranked votes to deserve representation. If $g = n$, for instance, then positional representation guarantees only that every alternative is ranked by some metric in $K$ at least as high as its lowest ranking. By the pigeonhole principle, it will always be possible to satisfy this specific guarantee with $|K| = 1$. At the other extreme, if $g = 1$, then we must have that $K = N$ in order to satisfy positional representation.

As a concrete example, suppose that we have $n = 100$ metrics and $g = 10$. If alternative $a$ is ranked first by exactly 23 metrics in $N$, then Definition 1 requires that $a$ is ranked first by at least two metrics in $K$. Similarly, if alternative $a$ is ranked in one of the top two places by exactly 76 metrics in $N$, then Definition 1 also requires that alternative $a$ is ranked in the top two by at least 7 metrics in $K$.

## 2.1 Lower Bound

In this section, we first provide a lower bound on the size of $K$ needed to satisfy positional representation. We then give an algorithm that returns a solution satisfying positional representation where $|K|$ is at most a logarithmic factor larger than the lower bound.

Observe that any $K$ satisfying positional representation for group size $g$ must have size $|K| \geq \lfloor \frac{n}{g} \rfloor$. This is because every alternative $a$ must be ranked in the top $m$ by all $n$ metrics, and therefore Definition 1 requires that $a$ is ranked in the top $m$ by at least $\lfloor \frac{n}{g} \rfloor$ metrics in $K$. Naturally, we might hope that for any $g$ and $N$, we can always find $K \subseteq N$ such that $K$ satisfies positional representation for group size $g$ and $|K| = \lfloor \frac{n}{g} \rfloor$. Unfortunately, we show this is impossible in the following example.

| $b_1$ | $b_2$ | $b_3$ | $b_4$ |
|-------|-------|-------|-------|
| $x$ | $x$ | $w$ | $w$ |
| $y$ | $z$ | $y$ | $z$ |
| $u$ | $v$ | $v$ | $u$ |
| $z$ | $y$ | $z$ | $y$ |
| $v$ | $u$ | $u$ | $v$ |
| $w$ | $w$ | $x$ | $x$ |

Table 2: Example where positional representation for group size $g = 2$ is impossible with $|K| = \frac{n}{g}$. Each metric in $\{b_1, b_2, b_3, b_4\}$ has preference ordering among the alternatives $\{u, v, w, x, y, z\}$ corresponding to that metric's column. Each color needs to be represented in $K$.

In this example, a set $K$ satisfies positional representation for group size $g = 2$ if and only if every color appears in $K$. However, there is no such subset $K \subset \{b_1, b_2, b_3, b_4\}$ where $|K| \leq 2 = \frac{n}{g}$, and therefore any $K$ satisfying positional representation for $g = 2$ must have $|K| \geq 3$. More generally, we show the following worst-case lower bound on the number of metrics needed to guarantee positional representation.

**Theorem 2** (Proof in Appendix F). *For every $g \geq 2$, there exists $\boldsymbol{\sigma}_N$ such that no subset $K \subseteq N$ satisfies positional representation for group size $g$ with size $|K| \leq \Omega\left( \frac{\frac{n}{g} \log(m)}{\log\left(\frac{n}{g} \log(m)\right)} \right)$.*

*Proof Sketch.* We will construct a preference profile $\boldsymbol{\sigma}_N$ where $K$ must be large to satisfy positional representation. First, we enumerate all possible subsets of $N$ of size $g$ as $\{G_1, ..., G_{\binom{n}{g}}\}$. We then construct a preference profile $\boldsymbol{\sigma}_N$ such that $\sigma_{ir} = a_r$ if $i \in G_r$ and $\sigma_{ir} = b_r$ if $i \notin G_r$ where $a_r, b_r$ are distinct alternatives for all $r \in \binom{n}{g}$. By this construction, for every $r \leq \binom{n}{g}$, a subset $K$ satisfying Equation (2) for alternative $a_r$ must include at least one metric from $G_r$. Therefore, $K$ must include at least one metric from every subset of size $g$ of $N$, which means $K$ must have size at least $n - g + 1$. We then show that $n - g + 1$ satisfies the desired bound. $\square$

Although achieving $|K| = \lfloor n/g \rfloor$ is not always possible, we still want to efficiently find a $K$ that satisfies positional representation for group size $g$ and contains relatively few metrics. We next present Algorithm 1, a polynomial time greedy algorithm that finds such a $K$ with $|K| \leq \frac{n}{g} \log(m)$.

## 2.2 Algorithm

We first give a high-level overview of the algorithm. The algorithm iterates through every element of the preference profile row by row. As it does so, it keeps track of how many times each alternative $j$ has shown up. Whenever $j$ has shown up $g$ times, the algorithm colors the last $g$ entries of $j$ with a new color and resets the counter for alternative $j$. Table 2 provides an example of the coloring at the end of this procedure. Note that if $n/g$ is not integral, not all of the elements will be colored.

After completing this process, the algorithm greedily selects metrics to include in the subset $K$ based on the number of colored alternatives in each metric's column that are not included in a previously selected metric. Specifically, the algorithm will select the first metric from the set of metrics that have

the most colored elements. The second metric is selected from the set of metrics that have the most *new* colors, and so on. This process continues until there are no new colors remaining among the unselected metrics, at which point the algorithm returns the set of selected metrics.

---

**Algorithm 1** Greedy (pseudo-code)

---

**Require:** Preference profile $\boldsymbol{\sigma}_N$, group size $g$
 1: **while** there exist alternatives with at least $g$ uncolored instances **do**
 2:     Choose an alternative $a$ with at least $g$ uncolored instances
 3:     Color the highest $g$ uncolored instances of $a$ with a new color (breaking ties arbitrarily)
 4: **end while**
 5: Initialize $K \leftarrow \emptyset$
 6: Let $C$ be the set of colors used
 7: **while** $C$ is nonempty **do**
 8:     Choose a metric $i \in N \setminus K$ that covers the most colors in $C$
 9:     Add $i$ to $K$
10:     Remove the colors that $i$ covers from $C$
11: **end while**
12: **return** $K$

---

The full algorithm is presented in Appendix D. Theorem 3 gives the formal bound for Algorithm 1.

**Theorem 3** (Proof in Appendix E). *For any preference profile $\boldsymbol{\sigma}_N$ and any group size $g$, Algorithm 1 terminates in polynomial time and returns a subset $K$ with $|K| \leq O(\frac{n}{g}\log(m))$ which satisfies positional representation for group size $g$.*

*Proof sketch.* The key idea of the proof is to keep track of the number of distinct colors that $K$ does not yet cover after iteration $t$ of the loop on Line 7 of Algorithm 1. Denote this quantity $Q_t$. By construction, the number of colors at the beginning of the loop is $Q_0 \leq \frac{mn}{g}$. The algorithm terminates at the smallest time $t$ where $Q_t = 0$. We first show that for each round of the loop, the metric $i \in N \setminus K$ that covers the most remaining colors in $C$ must cover at least $\frac{Q_t}{n/g}$ colors. Therefore, for all $t$, we have $Q_{t+1} \leq Q_t \left(1 - \frac{g}{n}\right)$. We also know that if $Q_t \leq n/g$, then $Q_{t+1} \leq Q_t - 1$. Combining these two equations, we show the desired result that $Q_t = 0$ for $t \geq (n/g + 1)\log(m)$. $\square$

Because any $K$ satisfying positional representation must have size at least $\lfloor n/g \rfloor$, Algorithm 1 selects no more than a $\log(m)$ factor more metrics than the smallest number needed to satisfy positional representation, For any $\boldsymbol{\sigma}_N$. For a given $\boldsymbol{\sigma}_N$, we can find the minimum number of metrics needed to satisfy positional representation using an integer program (see Appendix A); however this is not guaranteed to run in polynomial time.

## 3  Positional proportionality

While positional representation guarantees that each alternative gets sufficient representation in the subset of metrics chosen for each position cutoff, it does not prevent over-representation. For example, if an alternative $a$ is ranked in the top 10 in $\boldsymbol{\sigma}_N$ exactly $g$ times, then in order to satisfy positional representation with parameter $g$, $a$ must be ranked in the top 10 in $\boldsymbol{\sigma}_K$ at least once. However, there's no upper bound on how many times $a$ can be ranked in the top 10 — it could be possible to satisfy positional representation for this instance and have $a$ ranked in the top 10 by every metric in $K$.

In this section, we define a notion of proportionality which prevents both under-representation and over-representation. This notion, positional proportionality, is especially useful for recovering summary information such as the fraction of metrics which rank an alternative in the top half. We will show that if $K$ satisfies positional proportionality for $\boldsymbol{\sigma}_N$, then any positional scoring rule evaluated on $\boldsymbol{\sigma}_K$ is a good approximation for the same scoring rule evaluated on $\boldsymbol{\sigma}_N$.

Informally, positional proportionality guarantees that for every alternative and position cutoff, $K$ preserves the fraction of times that alternative is ranked above that position cutoff within an additive error $\epsilon$. From this guarantee, we can also recover the fraction of times that each alternative is ranked at each specific position within an additive error. The formal definition for positional proportionality follows below.

**Definition 4.** *A subset $K$ satisfies $\epsilon$-positional proportionality for $\epsilon \geq 0$ if for every alternative $a \in A$ and every $r \in [m]$, the fraction of metrics that rank $a$ in the top $r$ in $N$ is within $\epsilon$ of the fraction of metrics that rank $a$ in the top $r$ in $K$. Formally, for all $a \in A$ and $r \in [m]$,*

$$\left| \frac{C(N, r, a)}{|N|} - \frac{C(K, r, a)}{|K|} \right| \leq \epsilon. \tag{2}$$

Note that, unlike positional representation, positional proportionality is not parameterized by a group size $g$, but rather by an error $\epsilon$. As in positional representation, the choice of $\epsilon$ trades off the accuracy guarantee for proportionality with the minimum size of $K$ necessary. As $\epsilon$ increases, the size $|K|$ decreases, but the error in how well $K$ captures $N$ for each alternative and position cutoff may increase. As a concrete example of positional proportionality, suppose that we have $n = 100$ metrics and use parameter $\epsilon = \frac{1}{25}$. Further suppose that $a$ is ranked first by exactly 20 metrics in $N$. Then Definition 4 requires that the fraction of metrics in $K$ that rank $a$ first is between $\frac{4}{25}$ and $\frac{6}{25}$. Definition 4 further requires that approximate proportionality holds for every alternative at every position cutoff.

While both positional representation and positional proportionality enforce ways that $K$ must be representative of $N$, neither property implies the other, and neither is weakly easier to satisfy. In particular, positional representation strictly prevents under-representation, while positional proportionality approximately prevents both under-representation and over-representation. In the worst-case (and, we expect, in the typical case), the minimum number of metrics needed to satisfy positional representation with parameter $g$ is less than the minimum number of metrics needed to satisfy positional proportionality with $\epsilon = g/n$. However, this is not always the case. For instance, suppose that every metric in $N$ has the exact same ranking over alternatives. Then positional proportionality can be satisfied with $|K| = 1$ for any $\epsilon \geq 0$ by choosing an arbitrary metric. However, for any $g \leq |N|/2$, positional representation cannot be satisfied with less than $|K| = 2$. Intuitively, this is because positional proportionality gives a fractional guarantee, while positional representation gives an absolute guarantee, which sometimes allows positional proportionality to be more efficient in its information aggregation.

### 3.1 Upper and Lower Bounds

In this section, we present upper and lower bounds on the number of metrics necessary to guarantee positional proportionality. First, we show that for any preference profile $\boldsymbol{\sigma}_N$, there always exists a set $|K|$ with size $|K| = O\left(\frac{1}{\epsilon^2} \log(m)\right)$ that satisfies positional proportionality.

**Theorem 5** (Proof in Appendix G). *For every preference profile $\boldsymbol{\sigma}_N$ and $\epsilon \geq 0$, there exists $K \subset N$ with $|K| \leq \frac{1}{\epsilon^2} \log(2m)$ that satisfies positional proportionality.*

We next show that there exist preference profiles $\boldsymbol{\sigma}_N$ for which there is no $K$ with size less than $\Omega(\frac{1}{\epsilon^2} \log(m))$ that satisfies positional proportionality. Importantly, this shows that the result of Theorem 5 is tight up to constant factors.

**Theorem 6** (Proof in Appendix H). *For any $\epsilon \leq 1/24$, there exist $\boldsymbol{\sigma}_N$ such that no $K \subseteq N$ with $|K| \leq \frac{1}{288\epsilon^2} \log(m)$ satisfies $\epsilon$-positional proportionality.*

*Proof sketch.* We prove this result by designing a random $\boldsymbol{\sigma}_N$ such that with positive probability, there is no $K \subseteq N$ with $|K| \leq \frac{1}{288\epsilon^2} \log(m)$ that satisfies $\epsilon$-positional proportionality. We construct the random $\boldsymbol{\sigma}_N$ as follows. For every metric $i$ and every $j \in [m/2]$, with probability $1/2$ we will have alternative $2j$ ranked in position $2j$ and alternative $2j + 1$ in alternative $2j + 1$, and with probability $1/2$ we will have alternative $2j + 1$ ranked in position $2j$ and alternative $2j$ ranked in position $2j + 1$. This is done independently for all metrics $i$ and all $j$. For each fixed $K \subseteq N$ with $|K| \leq \frac{1}{288\epsilon^2} \log(m)$, we upper bound the probability that $K$ satisfies $\epsilon$-positional proportionality to be exponentially small. Intuitively, $K$ has an exponentially small probability of satisfying $\epsilon$-positional proportionality because the randomly assigned rankings must be approximately evenly distributed for all $j \in [\frac{m}{2}]$ simultaneously, and independence implies this has small probability. We formally show this using an inverse version of Hoeffding's Inequality. After bounding the probability of any fixed $K$ satisfying $\epsilon$-positional proportionality, a union bound gives that with positive probability, no such $K$ satisfies $\epsilon$-positional proportionality. This proves there must exist a profile $\boldsymbol{\sigma}_N$ such that no $K$ with $|K| \leq \frac{1}{288\epsilon^2} \log(m)$ satisfies $\epsilon$-positional proportionality. $\square$

As with positional representation, we can find the smallest $K$ that satisfies positional proportionality for a given instance using an integer program (see Equation (4) in Appendix A).

## 3.2 Approximating Scoring Rules

A nice feature of positional proportionality is that it allows us to approximate scoring rules evaluated on $\boldsymbol{\sigma}_N$ using only $\boldsymbol{\sigma}_K$. Informally, a scoring rule such as Borda count aggregates multiple rankings into a single ranking by assigning scores to each alternative based on its position in each of the original rankings [Young, 1975]. Because positional proportionality approximates the frequency at which an alternative $a$ is ranked above a cutoff $r$ up to an $\epsilon$ additive error, positional proportionality also approximates the frequency at which $a$ is ranked at exactly position $r$ up to a $2\epsilon$ additive error. This information in turn allows us to estimate the result of any scoring rule evaluated on $\boldsymbol{\sigma}_N$, which is especially helpful when $\boldsymbol{\sigma}_N$ is an intermediary of another computation, such as when $\boldsymbol{\sigma}_N$ is being used to decide a single winning alternative or a single meta-ranking.

Formally, a scoring rule has an associated score vector $s \in \mathbb{R}^m$, where $s_1 \geq ... \geq s_m$. It is without loss of generality to normalize so that $s_1 = 1$ and $s_m = 0$. When using a scoring rule to aggregate rankings, each metric awards $s_r$ points to the alternative that is ranked in position $r$, which results in each alternative having an average score of $f_s(a, \boldsymbol{\sigma}_N) := \frac{1}{|N|} \sum_{i \in N} s_{\sigma_i(a)}$. In Theorem 7, we show that given any scoring rule and any $K$ satisfying $\epsilon$-positional proportionality, every alternative has approximately the same average score in $\boldsymbol{\sigma}_K$ as in $\boldsymbol{\sigma}_N$.

**Theorem 7** (Proof in Appendix I). *If a subset $K$ satisfies $\epsilon$-positional proportionality, then for every scoring rule with score vector $s$ and every alternative $a \in A$, $|f_s(a, \boldsymbol{\sigma}_N) - f_s(a, \boldsymbol{\sigma}_K)| \leq \epsilon$.*

# 4 Generalizations

In previous sections, our goal has been to choose a subset of metrics that preserves rank information from the original set. However, there may be other types of information we would like to preserve instead of or in addition to rank information. For example, perhaps the metrics fall into different categories, and we want to include sufficiently many metrics of each category. It turns out that we can generalize both positional representation and positional proportionality to settings like this.

Formally, suppose we have a collection of $\gamma$ groups of metrics $\mathcal{G} = \{G_i\}_{i=1}^{\gamma}$ where $G_i \subseteq N$. Our goal is to choose a $K$ that represents every $G_i \in \mathcal{G}$. In the following two definitions, we generalize both positional representation and positional proportionality to this setting.

**Definition 8.** *For a given $N$ and collection of groups $\mathcal{G}$, a subset $K \subseteq N$ satisfies* generalized representation *for group size $g$ if for every $G_i \in \mathcal{G}$, $|K \cap G_i| \geq \left\lfloor \frac{|G_i|}{g} \right\rfloor$.*

**Definition 9.** *For a given $N$ and collection of groups $\mathcal{G}$, a subset $K \subseteq N$ satisfies $\epsilon$-generalized proportionality for $\epsilon \geq 0$ if for every $G_i \in \mathcal{G}$, $\left| \frac{|G_i|}{|N|} - \frac{|K \cap G_i|}{|K|} \right| \leq \epsilon$.*

Note that positional representation (Definition 1) and positional proportionality (Definition 4) are special cases of Definitions 8 and 9 for a specific choice of $\mathcal{G}$ that depends on $\boldsymbol{\sigma}_N$. Specifically, given $\boldsymbol{\sigma}_N$, we can construct $\mathcal{G}$ as follows. Let $\gamma = m^2$ and let $\mathcal{G} = \{G_{ar}\}$ where for each $a \in A$ and $r \in [m]$ we define $G_{ar} := \{i \in N : \sigma_i(a) \leq r\}$. In other words, for every $a \in A$ and $r \in [m]$, there is one group in $\mathcal{G}$ that corresponds to all of the metrics that rank $a$ in the top $r$ positions. By construction, any subset $K$ that satisfies generalized representation/proportionality for this choice of $\mathcal{G}$ will also satisfy positional representation/proportionality.

While the groups in Definitions 8 and 9 can be based on $\boldsymbol{\sigma}_N$, they need not be. Definitions 8 and 9 give us the freedom to define groups in whatever way is useful, which in turn allows us to specify which types of information to preserve. Below are some examples of groups we could define:

- Suppose some metrics are in English, some are in Chinese, and some are in Spanish. Then for each language, we could have a group in $\mathcal{G}$ corresponding to all metrics in that language.

- Suppose the metrics have a range of difficulty. Then for each difficulty level (e.g. very easy, easy, hard, very hard), we could have a group in $\mathcal{G}$ corresponding to all metrics of that difficulty level.

- Suppose we have ten experts who each believe a different subset of metrics are important. Then for each expert, we could have a group in $\mathcal{G}$ including all metrics that expert supports.

In Appendix B, we show how the lower and upper bounds of Theorems 2–6 can be generalized to give bounds for Definitions 8 and 9. In Appendix B, we also discuss the relationship between Definition 8 and set cover. Finally, we show that finding the smallest set that satisfies either Definition 9 or Definition 8 is NP-hard.

# 5 Empirical Case Studies

We demonstrate our proposed definitions and algorithms on three real-world case studies that involve selecting a subset of metrics for evaluation and decision making. To illustrate the wide potential applicability of our approach, we consider two case studies on evaluating LLM capabilities, and one case study on evaluating hospital quality. Each case study includes a full set of metrics, an existing subset of metrics currently deployed (e.g., an existing LITE benchmark), and a set of alternatives. Our experiments focus on two goals: *(i)* supplementing our theory by comparing the performance of our algorithms relative to the stated upper and lower bounds on real datasets, and *(ii)* demonstrating practical relevance by comparing against existing deployed subsets. An additional consideration is that a practitioner might want to keep an existing curated subset, so we also show that our algorithms can also be used to *augment* an existing subset. We summarize each case study below, and provide more details and code in the Supplemental Materials.

**Case Study 1: BIG-bench [Srivastava et al., 2022].**  We consider the problem of selecting a subset of $n = 141$ BIG-bench JSON metrics to include in a "lite" version. The existing BIG-bench Lite includes $k = 24$ JSON metrics, and was "designed to provide a canonical measure of model performance, while being far cheaper to evaluate than the full set." The alternatives consist of $m = 120$ LLMs from three model families.

**Case Study 2: HELM [Liang et al., 2022].**  We next consider the problem of selecting a subset of $n = 34$ scenarios available on HELM Classic for a Lite version. For this case study, we compare against the $k = 7$ metrics from HELM Lite that are from HELM Classic. The alternatives consist of $m = 67$ models that appeared on the HELM Classic leaderboard as of March, 2025.

**Case Study 3: Cal Hospital Compare [Cal Hospital Compare, 2025].**  Beyond LLMs, we also demonstrate how our methods can apply more widely through a case study on hospital quality evaluation. Cal Hospital Compare uses a subset of $k = 12$ hospital quality metrics selected from a full set of metrics collected by the Centers for Medicare and Medicaid Services (CMS). For the purposes of this illustration, we consider the problem of selecting a "representative" set of quality metrics from the $n = 50$ existing patient safety metrics available in the CMS Hospital Compare database. The alternatives consist of $m = 282$ hospitals in California.

## 5.1 Results

We now empirically evaluate the performance of the proposed algorithms for achieving positional representation (PR) and positional proportionality (PP). We evaluate each method by comparing the subset sizes $|K|$ achieved for each tolerance parameter (group size $g$ for PR, tolerance $\epsilon$ for PP). Smaller subsets are better.

**Positional Representation.**  We first evaluate the performance of Algorithm 1 (Greedy) relative to our theoretical upper and lower bounds, as well as the optimal integer programming solution.[3] Figure 1 shows that in practice, the greedy algorithm performs significantly better than the upper bound, but a gap remains relative to the optimal integer programming solution for small group sizes.

For all case studies, the greedy algorithm finds a subset smaller than the existing subset, while guaranteeing positional representation for a smaller group size $g$ than the existing subset guarantees (marked by the blue points in the bottom left quadrant). This suggests that if positional representation is important to a practitioner, it is possible to achieve it more efficiently than the existing subset.

In practice, the existing subset is often carefully curated. Thus, we also evaluate the performance of using our greedy and integer programming algorithms to optimally augment the existing subset for PR. Figure 1 shows that a subset optimally augmented using an integer program can perform better than the greedy algorithm for small group sizes. A computational advantage of augmenting the existing subset is that it reduces the free parameters of the integer programming problem. Thus, augmenting the existing subset could be a computationally practical approach that can beat the greedy algorithm in some cases, and is worth considering by practitioners.

**Positional Proportionality.**  For positional proportionality, we compare the integer programming solutions to the existing subsets. Figure 2 shows that for all datasets, the IP is able to find subsets of a smaller size than the existing subset that can guarantee PP at a lower $\epsilon$ tolerance. This again

---

[3]The optimal integer program is computationally feasible as our real instances have $n \leq 141$ and $m \leq 282$. Even when the IP is employed, Theorem 3 is directly useful as it upper bounds the size of the optimal solution.

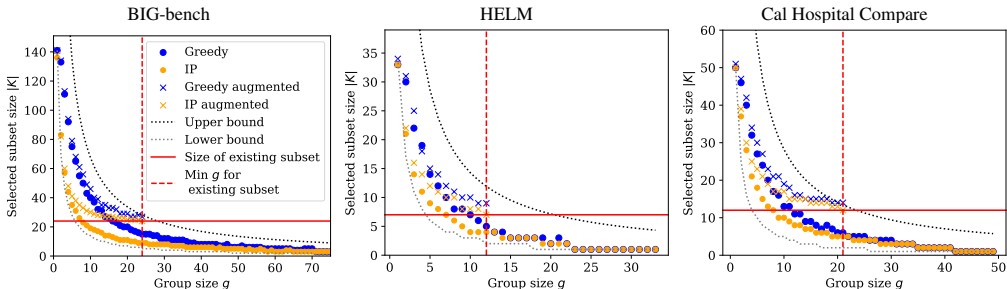

Figure 1: Results of running greedy and integer programming algorithms to achieve positional representation. The upper and lower bounds are from Theorems 3 and 2. The dashed red line marks the smallest $g$ for which the existing subset guarantees PR.

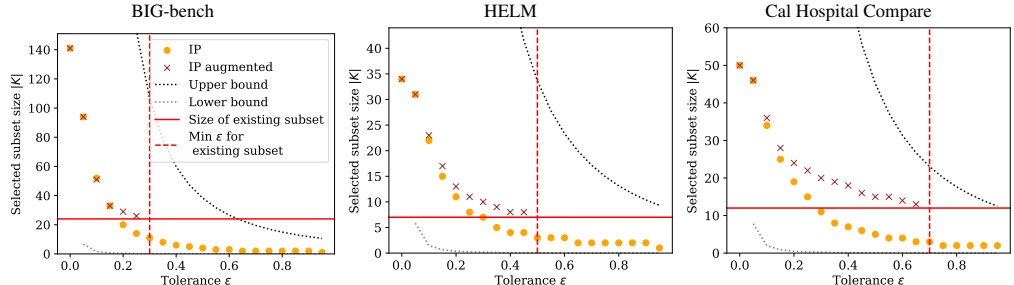

Figure 2: Results of running integer programming algorithms to achieve positional proportionality. The dashed red line marks the smallest $\epsilon$ for which the existing subset guarantees PP. The upper and lower bounds are from Theorems 5 and 6

suggests that there exist more efficient ways to achieve PP. As with PR, it is also possible to augment the existing subset to achieve PP. This was most apparent with Cal Hospital Compare, where it is possible to achieve a much smaller $\epsilon$ by adding less than five more metrics.

## 6 Discussion

We conclude by discussing limitations and future directions. In our work, we introduced two desirable properties that a subset of metrics should have in order to preserve rank information from the original set of metrics. One limitation is that it is not clear whether the subset chosen is good for evaluating new alternatives, especially if there is a distribution shift in the nature of alternatives over time (e.g., major advancements in LLMs). While we expect that the chosen subset of metrics will be reasonable at evaluating new alternatives in the short term, we recommend occasionally recomputing the subset as necessary to account for new metrics joining the overall set or paradigm shifts among the alternatives. As an open question, it would also be interesting to obtain theoretical guarantees about how well a selected subset of metrics generalizes to evaluating new alternatives.

Throughout our work, we attempt to curate a subset which is reflective of the overall set. However, if the overall set is biased or some types of metrics are overrepresented, we would expect that bias to be reflected in our subset as well. The purpose of our work is to select a good subset of metrics assuming the overall set captures what the user wants. This means our algorithms are agnostic to how and why the overall set of metrics was selected, which allows for tremendous flexibility, but also puts some onus on the user to make sure the overall set achieves the desired objective.

There are several other future directions of note. First, while we provide some case studies in Section 5, it would certainly be interesting to study other practical settings and the best choice of parameters $g$ and $\epsilon$ in each. In this work, we largely did not discuss what to do when there is missing data, i.e., if not all metrics rank all alternatives. While one natural approach is to assume all missing alternatives are tied for last, this is not the only viable approach, and we leave exploration in this direction to future work. Finally, certain metrics may have higher costs (e.g., running them may require more computational resources). It would be interesting to explore how to balance cost and usefulness of metrics when selecting a representative subset, especially if the user is budget-constrained.

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

# A    Integer Programs

In this section, we present integer programs which for a given instance $\boldsymbol{\sigma}_N$ give the smallest number of metrics necessary to guarantee either positional representation or positional proportionality

First, we define $S_{ra}$ to be the set of metrics which rank $a$ at least as high as $r$, i.e. $S_{ra} = \{i \in N : \sigma_i(a) \leq r\}$.

The IP for finding the minimum size set that satisfies positional representation is as follows:

$$
\begin{aligned}
\min \sum_{i \in N} & x_i \\
\text{s.t.} \sum_{i \in S_{ra}} x_i &\geq \left\lfloor \frac{C(N, r, a)}{g} \right\rfloor \quad \forall\, r \in [m], a \in A \\
x_i &\in \{0, 1\} \quad \forall\, i \in [n]
\end{aligned}
\tag{3}
$$

The IP for finding the minimum size subset that satisfies positional proportionality is as follows:

$$
\begin{aligned}
\min \sum_{i \in N} & x_i \\
\text{s.t.} \sum_{i \in S_{ra}} x_i &\geq \left( \frac{C(N, r, a)}{|N|} - \epsilon \right) \left( \sum_{i \in N} x_i \right) \quad \forall\, r \in [m], a \in A \\
\sum_{i \in S_{ra}} x_i &\leq \left( \frac{C(N, r, a)}{|N|} + \epsilon \right) \left( \sum_{i \in N} x_i \right) \quad \forall\, r \in [m], a \in A \\
x_i &\in \{0, 1\} \quad \forall\, i \in [n]
\end{aligned}
\tag{4}
$$

# B    General Versions and Set Cover

## B.1    Theoretical Results

Our main theoretical results from the previous two sections generalize to this setting. The main difference is that the $\log$ term in the theorem statement will now depend on the size of $\mathcal{G}$. Intuitively, if $\mathcal{G}$ is bigger, then more metrics are needed to guarantee representation for all $G \in \mathcal{G}$. We state the generalizations of the upper bounds in Theorems 10 and 11. Note that in the ranking setting, $|G| = m^2$, so Theorems 10 and 11 give a worse upper bound than Theorems 3 and 5. This is because for positional representation and positional proportionality, we have a better upper bound on the total size over all groups.

**Theorem 10.** *For any $\boldsymbol{\sigma}_N$, collection of groups $\mathcal{G}$, and group size $g$, the generalized greedy algorithm (Algorithm 2) terminates in polynomial time and returns a subset $K$ with $|K| \leq \frac{n}{g} \log\left(|\mathcal{G}|\right)$ which satisfies generalized representation for group size $g$ .*

*proof.* The proof of this result follows as in the proof of Theorem 3, except that instead of $Q_0 \leq m\alpha$ we have that $Q_0 \leq |\mathcal{G}|\alpha$. The rest of the recursion proof follows exactly the same to give the desired bound. □

**Theorem 11.** *For any $\boldsymbol{\sigma}_N$, collection of groups $\mathcal{G}$, and $\epsilon > 0$, there exists a subset $K$ with size $|K| \leq \frac{1}{2\epsilon^2} \log\left(4|\mathcal{G}|\right)$ that satisfies generalized proportionality.*

*proof.* The proof follows as in the proof of Theorem 5, except we now have a union bound over all $|\mathcal{G}|$ groups rather than all $m^2$ combinations of $a$ and $r$. □

Because positional representation and positional proportionality are special cases of general representation and general proportionality, the lower bounds of Theorems 2 and 6 also directly generalize to the general versions of these properties.

**B.2  Relationship to Set Cover**

General representation (Definition 8) is closely related to set cover, but has some key differences. In set cover, the input is a set of elements $U$ and a collection of subsets $\mathcal{C}$ where $C \subseteq U$ for all $C \in \mathcal{C}$. The goal is to find the smallest number of subsets from $\mathcal{C}$ whose union is equal to $U$. Similarly, in order to satisfy general representation, we need to find a subset of metrics which covers each group sufficiently many times. However, general representation differs from set cover in that in general representation, the number of times each $G$ (element) needs to be covered depends on the number of metrics that are in $G$. For example, if $|G| < g$, then $G$ does not need to be covered by $K$ at all. On the other hand, if $|G| = \ell g$, then $G$ needs to be covered at least $\ell$ times in $K$. Note that general representation also differs from the set multi-cover problem (where each element has a given number of times it must be covered) because in that problem, the number of times an element must be covered is not tied to its frequency [Chekuri et al., 2012, Hua et al., 2009].

More formally, suppose we index $\mathcal{G} = \{G_1, ..., G_\gamma\}$. Consider the set cover problem with $U = \{1, ..., \gamma\}$ and $\mathcal{C} = \{C_1, ..., C_n\}$, where $C_i = \{j \in [\gamma] : i \in G_j\}$. The goal of set cover for this choice of $U$ and $\mathcal{C}$ is to find the smallest $K \subseteq [N]$ such that for every $u \in U$, $|\{i \in K : u \in C_i\}| \geq 1$. Using the same notation, the goal of finding the smallest $K$ that satisfies general representation is equivalent to finding the smallest $K \subseteq N$ such that for every $u \in U$, $|\{i \in K : u \in C_i\}| \geq \left\lfloor \frac{|\{i \in N : u \in C_i\}|}{g} \right\rfloor$.

It is well-known that there exists an algorithm which achieves a $\ln(|\mathcal{C}|)$-approximation for set cover [Johnson, 1973, Chvatal, 1979]. We note that Theorem 10 is not subsumed by this result. First, observe that both these theorems guarantee an absolute bound on the number of benchmarks that are needed to satisfy general representation, rather than an approximation to the minimum number of benchmarks needed. In set cover, it is impossible to guarantee an absolute bound better than $|\mathcal{C}|$ – consider, for instance, the set cover instance where $\rfloor = \{\{u\} : u \in U\}$. Second, as mentioned earlier, general representation differs from set cover by requiring representation for an element based on the frequency that element appears.

We show that it is NP-hard to find the minimum size subset that satisfies either Definition 8 or Definition 9. The proofs of Theorems 12 and 13 can be found in Appendix K.1 and K.2 respectively.

**Theorem 12.** *The problem of finding the smallest set $K$ that satisfies generalized representation (Definition 8) is NP-hard.*

**Theorem 13.** *The problem of finding the smallest set $K$ that satisfies generalized proportionality (Definition 9) is NP-hard.*

# C  Additional Related Works

In the setting of JR, there are $n$ voters and $m$ candidates, and each voter indicates whether they approve of each candidate. Based on this approval information, the goal is to select a committee of size $k$ from the candidates. JR considers every cohesive coalition of voters, which is a group of size at least $n/k$ that all approve the same candidate. A committee then satisfies JR if at least one member of every such cohesive coalition approves of some candidate in the committee. Proportional Justified Representation (PJR) introduced by Sánchez-Fernández et al. [2017] extends JR by requiring further that that larger coalitions with more agreement will have more representation in the committee. Many other variations on justified representation have also been studied, including (but not limited to) Extended Justified Representation [Aziz et al., 2017], Full Justified Representation [Peters et al., 2021], Full Proportional Justified Representation [Kalayci et al., 2025]. Like JR, positional representation guarantees representation for every sufficiently large "coalition" of metrics that all rank an alternative $a$ above a position $r$, with proportionally more representation for larger coalitions. Unlike in JR, in positional representation there are no external voters; rather, the metrics serve as both the voters and the candidates. Furthermore (and also unlike JR), metrics do not indicate whether they approve of other metrics – instead, whether representation is deserved is based solely on the rankings.

There is also a line of work on proportionally fair clustering that uses similar notions of coalitions of size $n/k$ [Chen et al., 2019, Micha and Shah, 2020, Caragiannis et al., 2024]. These works differ from our setting in that there is no notion of ranking representation. Another major difference between these works and our setting is that our set of selected metrics must be a subset of all metrics, while clustering algorithms are generally able to choose any points as the centers of the cluster.

# D    Full Greedy Algorithm

During the algorithm, we assign labels (or "colors") to metrics. We use the set of natural numbers as labels, and let $c$ represent the lowest unused natural number. $C_i$ is the set of labels assigned to metric $i$. At any point in time, $S_j$ represents the set of metrics that have already approved alternative $j$ but have not yet been assigned a $j$-label.

---

**Algorithm 2** Greedy for positional representation

---

1: **Input:** Preference profile $\sigma_N$, group size $g$
2: Initialize $S_j \leftarrow \emptyset$ for all $j \in [m]$, $C_i \leftarrow \emptyset$ for all $i \in [n]$, and $c \leftarrow 1$
3: **for** each $r \in [m]$ **do**
4:     **for** each $i \in [n]$ **do**
5:         Let $j \leftarrow \sigma_{ir}$
6:         Add $i$ to $S_j$
7:         **if** $|S_j| = g$ **then**
8:             **for** each $i' \in S_j$ **do**
9:                 Add $c$ to $C_{i'}$
10:             **end for**
11:             $c \leftarrow c + 1$
12:             $S_j \leftarrow \emptyset$
13:         **end if**
14:     **end for**
15: **end for**
16: Let $C \leftarrow \{1, 2, \ldots, c - 1\}$, $K \leftarrow \emptyset$
17: **while** $C \neq \emptyset$ **do**
18:     Select $i \leftarrow \arg\max_{i'} |C_{i'}|$
19:     Add $i$ to $K$
20:     **for** each $x \in C_i$ **do**
21:         **for** each $i' \in [n]$ **do**
22:             **if** $x \in C_{i'}$ **then**
23:                 Remove $x$ from $C_{i'}$
24:             **end if**
25:         **end for**
26:         Remove $x$ from $C$
27:     **end for**
28: **end while**
29: **Return** $K$

---

# E    Proof of Theorem 3

*Proof of Theorem 3.* First, map each label to a unique color. Greedy adds one metric to the set $K$ in each iteration of the loop on Line 17 and this loop ends once all colors are covered. To bound the number of iterations, we analyze how many colors are not yet covered by $K$ at each step.

Let $\alpha = n/g$, and let $Q_t$ be the number of uncovered colors after $t$ iterations of the loop on Line 17 of Algorithm 3. Initially, $Q_0 = m\lfloor \frac{n}{g} \rfloor \leq m \cdot \alpha$. Each color is covered by exactly $g$ metrics and is covered at most once by each metric, so if there are strictly more than $s \cdot \alpha$ colors remaining for any integer $s \geq 1$, there must exist a metric covering at least $s + 1$ uncovered colors. In other words, if there are $Q_t$ colors remaining after iteration $t$, then the next metric chosen at iteration $t + 1$ must contain at least $\lceil Q_t/\alpha \rceil$ distinct colors. This gives the following recurrence:

$$Q_{t+1} \leq Q_t - \lceil Q_t/\alpha \rceil \leq Q_t \cdot (1 - 1/\alpha).$$

Because $Q_0 \leq m \cdot \alpha$, this means that

$$Q_t \leq m \cdot \alpha \cdot (1 - 1/\alpha)^t. \tag{5}$$

Once $Q_t \leq \alpha$, every metric chosen by the algorithm will still contain at least one new uncovered color, so for $Q_t \leq \alpha$ we have that

$$Q_{t+1} \leq Q_t - 1.$$

This means that once $Q_t \leq \alpha$, the loop will finish in at most $\alpha$ additional steps.

Now, we will upper bound the number of steps until $Q_t \leq \alpha$. By Equation (5), we have that $Q_t \leq \alpha$ for any $t$ satisfying

$$m \cdot \alpha \cdot (1 - 1/\alpha)^t \leq \alpha.$$

Solving and simplifying this equation gives that $Q_t \leq \alpha$ for any $t$ satisfying

$$t \geq \log(m)/\log(\alpha/(\alpha - 1)).$$

Note that

$$\log(\alpha/(\alpha - 1)) = \log(\alpha) - \log(\alpha - 1) = \int_{\alpha-1}^{\alpha} 1/x \, dx \geq 1/\alpha.$$

Combining the previous two equations, we have that $Q_t \leq \alpha$ for any $t$ satisfying

$$t \geq \alpha \cdot \log(m).$$

As we argued above, the algorithm will only add at most $\alpha$ more metrics once $Q_t \leq \alpha$, therefore the total number of metrics added to $K$ by Greedy is at most

$$\alpha + \alpha \cdot \log(m) = O\left(\frac{n}{g} \log(m)\right)$$

as desired. $\qquad \square$

## F    Proof of Theorem 2

*Proof of Theorem 2.* Fix any $g \geq 2$. Choose any $m$ and $n$ such that $\frac{n}{g} \geq 3$ is an integer and such that

$$m = 2\binom{n}{g}.$$

Construct $\boldsymbol{\sigma}_N$ as follows. First, enumerate all $\binom{n}{g}$ distinct subsets of $N$ of size $g$ as $\{G_1, ..., G_{\binom{n}{g}}\}$. We will assign 2 distinct alternatives to each of the first $\binom{n}{g}$ ranking places. Let $\{a_r, b_r\}$ be the alternatives assigned to rank $r$ for $r \in [1 : \binom{n}{g}]$. Let $\sigma_{ir} = a_r$ if $i \in G_r$ and $\sigma_{ir} = b_r$ if $i \notin G_r$. Therefore, alternative $a_r$ will be ranked $r$ by exactly $g$ metrics while alternative $b_r$ will be ranked $r$ by exactly $n - g$ metrics. This means that only alternatives in $\{a_r, b_r\}$ will be ranked in position $r$ for any of the metrics.

Now set the rest of the rankings (for positions $\binom{n}{g} + 1$ through $m$) of the metrics arbitrarily in any valid way. This will not matter for the rest of the proof.

Define $\alpha = n/g$. Next, we will show that no set of metrics can satisfy positional representation for group size $g$ in this example with $K$ having size less than $(\alpha - 1)g + 1$. Proof by contradiction. Suppose we have a set $K$ such that $|K| \leq (\alpha - 1)g$ and $K$ satisfies positional representation for group size $g$. Then there must be a set of $g$ metrics not included in $K$. Denote this set of $g$ metrics as $G$. Because we used every possible subset of $N$ for the permutations of $a_r, b_r$ in the first $\binom{n}{g}$ positions, there is some position $\hat{r}$ and some alternative $a_{\hat{r}}$ such that $a_{\hat{r}}$ is ranked exactly $\hat{r}$ in every metric in $G$ and such that $a_{\hat{r}}$ does not appear ranked in the top $\hat{r}$ in any metric not in $G$. In order for $K$ to satisfy positional representation for group size $g$, at least one of the rankings in $G$ must be included in $K$, which is a contradiction.

Therefore, we must have that any $K$ satisfying positional representation for group size $g$ must satisfy $|K| > (\alpha - 1)g$.

Furthermore,

$$\log(m) = \log\left(2\binom{n}{g}\right) \leq \log(2n^g) = g\log(n) + \log(2) \leq 2g\log(n).$$

The previous equation implies that $g \geq \frac{\log(m)}{2\log(n)}$. Using this on the third line below, we have that for any $K$ satisfying positional representation,

$$
\begin{aligned}
|K| &\geq (\alpha - 1)g \\
&= \left(\frac{n}{g} - 1\right)g \\
&\geq \frac{(n/g - 1)\log(m)}{2\log(n)} \\
&\geq \frac{\frac{n}{2g}\log(m)}{2\log(n)} \\
&= \frac{\frac{n}{g}\log(m)}{4\log(n)}.
\end{aligned} \tag{6}
$$

The last step is to upper bound $\log(n)$. By construction,

$$
m = 2\binom{n}{g} \geq 2\left(\frac{n}{g}\right)^g = 2\alpha^{n/\alpha}.
$$

Taking the log of both sides,

$$
\log(m) \geq \log(2) + \frac{n}{\alpha}\log(\alpha),
$$

which simplifies to

$$
n \leq \frac{\alpha(\log(m) - \log(2))}{\log(\alpha)}.
$$

Taking a log of both sides again gives

$$
\log(n) \leq \log\left(\frac{\alpha(\log(m) - \log(2))}{\log(\alpha)}\right) \leq \log(\alpha\log(m)) = \log\left(\frac{n}{g}\log(m)\right).
$$

Combining this with Equation (6), we have the desired result that for any $K$ satisfying positional representation,

$$
|K| \geq \frac{\frac{n}{g}\log(m)}{4\log\left(\frac{n}{g}\log(m)\right)}.
$$

$\square$

## G    Proof of Theorem 5

We use the probabilistic method. Suppose we select $K$ by choosing exactly $\frac{1}{\epsilon^2}\log(2m)$ random metrics from $N$ one-by-one. Consider any fixed alternative $a$ and position $r$. Let $X_1, ..., X_{|K|}$ be indicator random variables where $X_i$ is 1 if $a$ is ranked in the top $r$ by the $i$th chosen metric in $K$. Then we have that $\mathbb{E}[X_i] = \frac{C(N,r,a)}{|N|}$ for all $i$. We will use the following version of Hoeffding's inequality for random variables chosen without Replacement

**Lemma 14** (Hoeffding's without replacement Bardenet and Maillard [2015]). *Let $X = (x_1, \ldots, x_N)$ be a finite population of $N$ real numbers, and let $X_1, \ldots, X_n$ be a random sample drawn* without replacement *from $X$. Define:*

$$
a = \min_{1 \leq i \leq N} x_i, \quad b = \max_{1 \leq i \leq N} x_i, \quad \mu = \frac{1}{N}\sum_{i=1}^{N} x_i.
$$

*Then, for all $\varepsilon > 0$,*

$$
\mathbb{P}\left(\left|\frac{1}{n}\sum_{i=1}^{n} X_i - \mu\right| \geq \varepsilon\right) \leq 2\exp\left(-\frac{2n\varepsilon^2}{(b-a)^2}\right).
$$

Applying Lemma 14, we have that

$$\Pr\left(\left|\frac{C(N,r,a)}{|N|} - \frac{C(K,r,a)}{|K|}\right| > \epsilon\right) = \Pr\left(\left|\frac{1}{|K|}\sum_{i=1}^{|K|}X_i - \frac{C(N,r,a)}{|N|}\right| > \epsilon\right)$$

$$\leq 2e^{-2|K|\epsilon^2}$$

$$= 2e^{-2\log(2m)}$$

$$= \frac{1}{2m^2}.$$

By a union bound over all $m^2$ combinations of $a$ and $r$, we then have that

$$\Pr\left(\exists a, r : \left|\frac{C(N,r,a)}{|N|} - \frac{C(K,r,a)}{|K|}\right| > \epsilon\right) \leq m^2 \cdot \frac{1}{2m^2} = 1/2.$$

We can then take the complement of the event above to observe that $K$ satisfies positional proportionality with probability at least $\frac{1}{2}$. This probability is positive, so there must exist a $K$ with $|K| \leq \frac{1}{\epsilon^2}\log(2m)$ that satisfies positional proportionality.

## H    Proof of Theorem 6

*Proof of Theorem 6.* We will prove this using the probabilistic method. We will show that there exists a random generation process for $\boldsymbol{\sigma}_N$ such that with non-0 probability, no $K \subseteq N$ with $|K| \leq \frac{1}{288\epsilon^2}\log(m)$ satisfies $\epsilon$-positional proportionality.

First, we will choose $n$ and $m$ such that $m$ is even and $n$ and $m$ are sufficiently large so that the following three equations hold:

$$\frac{\log(m)\log(n+1)}{288\epsilon^2} - \frac{\sqrt{m}}{60} < -\log(2) \tag{7}$$

$$\frac{\log(m)\log(n+1)}{288\epsilon^2} + \log(m) - 2n\epsilon^2 < -\log(2) \tag{8}$$

$$1/\epsilon \leq \log_2\left(30\sqrt{m}\right). \tag{9}$$

Consider the following random generation process for $\boldsymbol{\sigma}_N$. For metric $i \in [n]$, the ranking for metrics $i$ will be generated as follows. With probability $1/2$, metric $i$ will rank alternative 1 in the first position and alternative 2 in the second position, and with probability $1/2$ metric $i$ will rank alternative 2 in the first position and alternative 1 in the second position. Repeat this process for all subsequent pairs of odd/even positions. So for all $j \in [\frac{m}{2}]$, with probability $1/2$ metric $i$ will rank alternative $2j$ in the $2j$ position and alternative $2j + 1$ in the $2j + 1$ position, and with probability $1/2$ metric $i$ will rank alternative $2j + 1$ in the $2j$ position and alternative $2j$ in the $2j + 1$ position.

We formalize this process as follows. For every $j \in [\frac{m}{2}]$ and $i \in [1 : n]$, let $X_{ij} \sim$ Bernoulli$(0.5)$. If $X_{ij} = 0$, then metric $i$ ranks alternative $2j$ in the $2j$ position and alternative $2j + 1$ in the $2j + 1$ position. If $X_{ij} = 1$, then metric $i$ ranks alternative $2j + 1$ in the $2j$ position and alternative $2j$ in the $2j + 1$ position.

We will now show that with positive probability, no $K \subseteq N$ with $|K| \leq \frac{1}{288\epsilon^2}\log(m)$ will satisfy $\epsilon$-positional proportionality for the random $\boldsymbol{\sigma}_N$ generated as described above.

First, we define an event $E$, which corresponds to the event that for all $j$, approximately $1/2$ of the metrics rank alternative $2j$ above alternative $2j + 1$ and approximately $1/2$ of the metrics rank alternative $2j + 1$ above alternative $2j$. Formally, define

$$E := \left\{\forall j \in [\tfrac{m}{2}], \left|\frac{1}{n}\sum_{i=1}^{n}(X_{ij} - 0.5)\right| \leq \epsilon\right\}.$$

By Hoeffding's inequality and a union bound,

$$\Pr(\neg E) = \Pr\left(\exists j \in [\tfrac{m}{2}], \left|\frac{1}{n}\sum_{i=1}^{n}(X_{ij} - 0.5)\right| > \epsilon\right)$$

$$\leq \sum_{j=1}^{m/2}\Pr\left(\left|\frac{1}{n}\sum_{i=1}^{n}(X_{ij} - 0.5)\right| > \epsilon\right) \qquad\qquad \text{[Union Bound]}$$

$$\leq me^{-2n\epsilon^2}. \qquad\qquad\qquad\qquad\qquad \text{[Hoeffding's Inequality]} \qquad (10)$$

Now consider any subset $K \subseteq N$ with $|K| \leq \frac{1}{288\epsilon^2}\log(m)$. We will lower bound the probability that this $K$ does not satisfy positional proportionality.

$\Pr(K \text{ does not satisfy } \epsilon\text{-positional proportionality})$

$$= \Pr\left(\exists a \in A, r \in [m] : \left|\frac{C(N,r,a)}{n} - \frac{C(K,r,a)}{|K|}\right| > \epsilon\right)$$

$$= \Pr\left(\exists j \in [\tfrac{m}{2}] : \left|\frac{1}{|K|}\sum_{i\in K}X_{ij} - \frac{1}{n}\sum_{i=1}^{n}X_{ij}\right| > \epsilon\right)$$

$$\geq \Pr\left(\left\{\forall j \in [\tfrac{m}{2}], \left|\frac{1}{n}\sum_{i=1}^{n}X_{ij} - 0.5\right| \leq \epsilon\right\} \bigcap \left\{\exists j \in [\tfrac{m}{2}] : \left|\frac{1}{|K|}\sum_{i\in K}(X_{ij} - 0.5)\right| > 2\epsilon\right\}\right) \quad [\triangle\text{-ineq.}]$$

$$= \Pr\left(E \bigcap \left\{\exists j \in [\tfrac{m}{2}] : \left|\frac{1}{|K|}\sum_{i\in K}(X_{ij} - 0.5)\right| > 2\epsilon\right\}\right)$$

$$\geq \Pr\left(\exists j \in [\tfrac{m}{2}] : \left|\frac{1}{|K|}\sum_{i\in K}(X_{ij} - 0.5)\right| > 2\epsilon\right) - \Pr(\neg E) \qquad [\Pr(A\cap B) \geq \Pr(A) - \Pr(\neg B)]$$

$$\geq \Pr\left(\exists j \in [\tfrac{m}{2}] : \left|\frac{1}{|K|}\sum_{i\in K}(X_{ij} - 0.5)\right| > 2\epsilon\right) - me^{-2n\epsilon^2} \qquad [\text{Equation (10)}]$$

$$= 1 - \Pr\left(\forall j \in [\tfrac{m}{2}] : \left|\frac{1}{|K|}\sum_{i\in K}(X_{ij} - 0.5)\right| \leq 2\epsilon\right) - me^{-2n\epsilon^2}$$

$$= 1 - \prod_{j=1}^{m/2}\Pr\left(\left|\frac{1}{|K|}\sum_{i\in K}(X_{ij} - 0.5)\right| \leq 2\epsilon\right) - me^{-2n\epsilon^2} \qquad [\text{Ind. of } X_{ij}]$$

$$\geq 1 - \left(1 - \frac{1}{30\sqrt{m}}\right)^{m/2} - me^{-2n\epsilon^2} \qquad [\text{Lemma (15)}]$$

$$\geq 1 - \left(e^{-\frac{1}{30\sqrt{m}}}\right)^{m/2} - me^{-2n\epsilon^2} \qquad [1 + x \leq e^x]$$

$$= 1 - e^{-\frac{\sqrt{m}}{60}} - e^{\log(m) - 2n\epsilon^2}. \qquad (11)$$

Using the above equation, we can bound the probability that there exists a $|K|$ with size less than $\frac{1}{288\epsilon^2} \log(m)$ that satisfies positional proportionality as follows:

$$\Pr(\exists K \subseteq N \text{ s.t. } |K| \leq \frac{1}{288\epsilon^2} \log(m) \text{ and } K \text{ satisfies } \epsilon\text{-positional proportionality})$$

$$\leq \sum_{K \subseteq N, |K| \leq \frac{1}{288\epsilon^2} \log(m)} \Pr(K \text{ satisfies } \epsilon\text{-positional proportionality}) \qquad \text{[Union Bound]}$$

$$\leq \sum_{K \subseteq N, |K| \leq \frac{1}{288\epsilon^2} \log(m)} \left( e^{-\frac{\sqrt{m}}{60}} + e^{\log(m) - 2n\epsilon^2} \right) \qquad \text{[Eq. (11)]}$$

$$\leq (n+1)^{\frac{1}{288\epsilon^2} \log(m)} \left( e^{-\frac{\sqrt{m}}{60}} + e^{\log(m) - 2n\epsilon^2} \right)$$

$$= e^{\frac{\log(m) \log(n+1)}{288\epsilon^2}} \left( e^{-\frac{\sqrt{m}}{60}} + e^{\log(m) - 2n\epsilon^2} \right)$$

$$= \text{Exp}\left( \frac{\log(m) \log(n+1)}{288\epsilon^2} - \frac{\sqrt{m}}{60} \right) + \text{Exp}\left( \frac{\log(m) \log(n+1)}{288\epsilon^2} + \log(m) - 2n\epsilon^2 \right)$$

$$< \text{Exp}\left( -\log(2) \right) + \text{Exp}\left( -\log(2) \right) \qquad \text{[Eqs (7) and (8)]}$$

$$= 1/2 + 1/2$$

$$= 1.$$

Therefore, we have shown that with positive probability, there will be no $K$ with size $|K| \leq \frac{1}{288\epsilon^2} \log(m)$ that satisfies $\epsilon$-positional proportionality. Finally, we can conclude that there *must* exist some ranking $\boldsymbol{\sigma}_N$ such that no $K$ with size $|K| \leq \frac{1}{288\epsilon^2} \log(m)$ satisfies $\epsilon$-positional proportionality.

In the equations above, we used the following inverse of Hoeffding's inequality.

**Lemma 15.** *Using the notation and setting of the proof of Theorem 6 above, for any $K$ satisfying $|K| \leq \frac{1}{288\epsilon^2} \log(m)$ and any $j \in [\frac{m}{2}]$,*

$$\Pr\left( \left| \frac{1}{|K|} \sum_{i \in K} (X_{ij} - 0.5) \right| > 2\epsilon \right) \geq \frac{1}{30\sqrt{m}}.$$

*Proof of Lemma 15.* We will prove this separately for small $|K|$ and for large $|K|$.

If $|K| \leq 1/\epsilon$, then

$$\Pr\left( \left| \frac{1}{|K|} \sum_{i \in K} (X_{ij} - 0.5) \right| > 2\epsilon \right) \geq \Pr\left( \frac{1}{|K|} \sum_{i \in K} X_{ij} = 1 \right) \qquad [\epsilon \leq 1/24]$$

$$= 2^{-|K|}$$

$$\geq 2^{-1/\epsilon}$$

$$\geq \frac{1}{30\sqrt{m}}, \qquad \text{Equation (9)}$$

which is the desired result.

To prove the desired result when $|K| > 1/\epsilon$, we will use the following proposition from Matoušek and Vondrák [2001]:

**Proposition 16** (Proposition 7.3.2 of Matoušek and Vondrák [2001])**.** *Let $X_1, ..., X_n$ be independent Bernoulli random variables with probability $1/2$ of being 0 or 1. Let $X = X_1 + ... + X_n$. Then for any integer $t \in [0, n/8]$,*

$$\Pr(X \geq n/2 + t] \geq 1/30 e^{-16t^2/n}.$$

We can apply this proposition to our problem in the following way. If $|K| > 1/\epsilon$, then

$$\Pr\left(\left|\frac{1}{|K|}\sum_{i\in K}(X_{ij}-0.5)\right| > 2\epsilon\right)$$

$$= \Pr\left(\left|\sum_{i\in K}(X_{ij}-0.5)\right| > 2|K|\epsilon\right)$$

$$\geq \Pr\left(\left|\sum_{i\in K}(X_{ij}-0.5)\right| \geq 2|K|\epsilon + 0.5\right)$$

$$\geq \Pr\left(\left|\sum_{i\in K}(X_{ij}-0.5)\right| \geq 3|K|\epsilon\right) \qquad\qquad [|K| > 1/\epsilon]$$

$$\geq \Pr\left(\sum_{i\in K}X_{ij} \geq \frac{|K|}{2} + 3|K|\epsilon\right)$$

$$\geq \frac{1}{30}e^{-16(3|K|\epsilon)^2/(|K|)} \qquad\qquad [\text{Prop } 16 \text{ with } t = 3|K|\epsilon]$$

$$= \frac{1}{30}e^{-144|K|\epsilon^2}$$

$$\geq \frac{1}{30}e^{-144\frac{1}{288\epsilon^2}\log(m)\epsilon^2}$$

$$= \frac{1}{30}e^{-\log(m)/2}$$

$$= \frac{1}{30\sqrt{m}}, \qquad\qquad\qquad\qquad\qquad\qquad\qquad (12)$$

which is the desired result. Note that we could apply Proposition 16 because $3|K|\epsilon \leq |K|/8$ for $\epsilon \leq 1/24$.

$\square$

$\square$

# I Proof of Theorem 7

*proof.* If a subset $K$ satisfies $\epsilon$ positional proportionality for set $N$, then

$$\left|\frac{C(N,r,a)}{n} - \frac{C(K,r,a)}{|K|}\right| \leq \epsilon \quad \forall r, a.$$

Consider a scoring rule with scoring vector $s$, i.e. that gives score $s_r$ to an alternative in position $r$. Define $\Delta_r = s_r - s_{r+1}$ (where $s_{m+1} = 0$). Then by definition of scoring rule, we have that

$$f_s(a, \sigma_N) = \frac{1}{n}\sum_{i=1}^{n} s_{\sigma_i(a)} = \frac{1}{n}\sum_{r=1}^{m} C(N,r,a)\Delta_r.$$

and

$$f_s(a, \sigma_K) = \frac{1}{|K|}\sum_{i\in K} s_{\sigma_i(a)} = \frac{1}{|K|}\sum_{r=1}^{m} C(K,r,a)\Delta_r.$$

Combining this with the first equation, we have that

$$|f_s(a, \sigma_N) - f_s(a, \sigma_K)| \leq \sum_{r=1}^{m} \left| \frac{C(N, r, a)}{n} - \frac{C(K, r, a)}{|K|} \right| \Delta_r$$

$$\leq \sum_{r=1}^{m} \epsilon \Delta_r$$

$$= \epsilon \sum_{r=1}^{m} (s_r - s_{r+1})$$

$$= \epsilon (s_1 - s_{m+1})$$

$$= \epsilon,$$

where the last line follows from the fact that WLOG $s_1 = 1$. □

## J   General Greedy Algorithm

---
**Algorithm 3** Greedy for General Representation
---
1: **Input:** Preference profile $\sigma_N$, collection of groups $\mathcal{G}$, group size $g$
2: Initialize $S \leftarrow \emptyset$, $C_i \leftarrow \emptyset$ for all $i \in [n]$, and $c \leftarrow 1$
3: **for** each $G \in \mathcal{G}$ **do**
4:     **for** each $i \in [n]$ **do**
5:         **if** $i \in G$ **then**
6:             Add $i$ to $S$
7:         **end if**
8:         **if** $|S| = g$ **then**
9:             **for** each $i' \in S$ **do**
10:                 Add $c$ to $C_{i'}$
11:             **end for**
12:             $c \leftarrow c + 1$
13:             $S \leftarrow \emptyset$
14:         **end if**
15:     **end for**
16: **end for**
17: Let $C \leftarrow \{1, 2, \ldots, c - 1\}$, $K \leftarrow \emptyset$
18: **while** $C \neq \emptyset$ **do**
19:     Select $i \leftarrow \arg\max_{i'} |C_{i'}|$
20:     Add $i$ to $K$
21:     **for** each $x \in C_i$ **do**
22:         **for** each $i' \in [n]$ **do**
23:             **if** $x \in C_{i'}$ **then**
24:                 Remove $x$ from $C_{i'}$
25:             **end if**
26:         **end for**
27:         Remove $x$ from $C$
28:     **end for**
29: **end while**
30: **Return** $K$
---

# K NP-Hardness Proofs

## K.1 Proof of Theorem 12

*proof.* Suppose we have an instance of set cover, i.e. a collection of sets $\mathcal{C} = \{C_1, ..., C_\kappa\}$ for which we want to find the smallest set that contains every distinct element in the sets of $\mathcal{C}$. Let $U = \bigcup_{C \in \mathcal{C}} C$. We want to map this to an instance of our problem, which consists of an $N$, $\mathcal{G}$, and $g$ for which we would like to find the smallest subset $K$ that satisfies general representation.

The key idea of the proof is to construct an instance of our problem where every group in $\mathcal{G}$ has the same size. Every element in $u \in U$ will have a corresponding group $G_u$ in $\mathcal{G}$. We start with each $G_u$ initialized to be the indices of the elements in $\mathcal{C}$ that contain $u$. However, this results in the $G_u$ having different sizes. We then add new tasks to $N$ that are only contained in a single group $G_u$ until all groups in $\mathcal{G}$ have the same size.

Formally, we construct $N$, $\mathcal{G}$, and $g$ as follows:

---
**Algorithm 4** Constructing $N, \mathcal{G}, g$
---
1: **Input:** $U, \{C_i\}_{i \in [\kappa]}$
2: $G_u \leftarrow \{i : u \in C_i\}$ for all $u \in U$
3: $g \leftarrow \max_u |G_u|$
4: $i \leftarrow \kappa$
5: **for** $u \in U$ **do**
6:     **while** $|G_u| < g$ **do**
7:         $i \leftarrow i + 1$
8:         Add $i$ to $G_u$
9:     **end while**
10: **end for**
11: **Return** $\mathcal{G} = \{G_u\}_{u \in U}$, $N = [i]$, and $g$

---

By construction, $|G_u| = g$ for all $u$.

We will now show that the set cover instance $(U, \mathcal{C})$ has a solution of size less than or equal to $k$ if and only if there exists a solution of size less than or equal to $k$ that satisfies general representation for $N, \mathcal{G}, g$. Suppose the set cover instance has a solution $\tilde{\mathcal{C}}$ with $|\tilde{\mathcal{C}}| \leq k$. Then this $\tilde{\mathcal{C}}$ covers every element in $U$ at least once. By construction, this $K = \{i \leq \kappa : C_i \in \tilde{\mathcal{C}}\}$ is a solution to the general representation problem, because general representation for group size $g$ where every group in $\mathcal{G}$ has size $g$ requires that $|G_u \cap K| \geq 1$ for all groups $G_u \in \mathcal{G}$.

For the opposite direction, suppose we have some $K \subseteq N$ with $|K| \leq k$ that satisfies general representation. Define $K'$ as in Algorithm 5.

---
**Algorithm 5** Constructing $K'$
---
1: **Input:** $K$
2: $K' \leftarrow K \cap [\kappa]$
3: **for** $i \in K \setminus [\kappa]$ **do**
4:     $u \leftarrow u$ such that $i \in G_u$
5:     $i' \leftarrow$ any element of $[\kappa]$ such that $u \in C_{i'}$
6:     **if** $i' \notin K'$ **then**
7:         $K' \leftarrow K' \cup \{i'\}$
8:     **end if**
9: **end for**
10: **Return** $K'$

---

Note that at the end of this algorithm, $|K'| \leq |K|$ because we never add an $i'$ to $K'$ unless there is a corresponding $i \in K \setminus [\kappa]$ that is not in $K'$. Furthermore, $K'$ still covers every group $G_u$, because the $i'$ selected in Line 5 covers the $G_u$ that was previously covered by $i$ in $K$. Note that such an $i'$ always exists because every $u \in U$ is in at least one subset in $\mathcal{C}$.

Therefore, we have that $K'$ still covers every group in $G_u$ and $|K'| \leq |K| \leq k$. This means that $\tilde{\mathcal{C}} = \{C_i : i \in K'\}$ is a solution to the set cover instance with size less than or equal to $k$. $\qquad\square$

## K.2 Proof of Theorem 13

*proof.* We will follow a similar proof structure to the proof of Theorem 1 in Natarajan [1995] on the NP-hardness of sparse approximate solutions to linear equations. As in Natarajan [1995], we will reduce from the problem of "exact cover by 3 sets".

Exact Cover by 3-Sets takes as input a set $S$ with $|S| = \tau$ and a collection $\mathcal{C} = \{C_1, ..., C_\kappa\}$ of subsets of $S$ such that $|C_i| = 3$ for all $i \leq \kappa$, and the goal is to determine whether or not there exists a subset of $\mathcal{C}$ (denoted $\tilde{\mathcal{C}}$) such that $\tilde{\mathcal{C}}$ covers every element in $S$ exactly once.

We will next describe how to map an instance of Exact Cover by 3-Sets to an instance of a decision version of our problem. Specifically, we will consider the problem where we are given $N, \mathcal{G}$, and $\epsilon$ and must decide whether there exists a subset $K \subseteq N$ of size $\tau/3$ that satisfies $\epsilon$-general proportionality.

Suppose we have an instance of Exact Cover by 3-Sets $(S, \mathcal{C})$. Let $\epsilon = \frac{1}{\tau}$ and let $\mathcal{G} = \{G_s\}_{s \in S}$ for $G_s$ constructed as follows. We will construct $G_s$ so that $|G_s| = 2\kappa$ for all $s \in S$. We will further maintain that for every $C_i$, we have $i \in G_s$ for all $s \in C_i$ and that every $i > \kappa$ appears in at most two groups $G_s$.

We formally construct $\mathcal{G}$ in Algorithm 6.

---

**Algorithm 6** Constructing $\mathcal{G}$

---

1: **Input:** $S, \mathcal{C} = \{C_i\}_{i \in [\kappa]}$
2: $G_s \leftarrow \{i : s \in C_i\}$ for all $s \in S$
3: $i \leftarrow \kappa$
4: **while** $\exists s \in S : |G_s| < 2\kappa$ **do**
5:    $i \leftarrow i + 1$
6:    **if** there is exactly one $s \in S$ such that $|G_s| < 2\kappa$ **then**
7:       $s \leftarrow s \in S$ such that $|G_s| < 2\kappa$
8:       Add $i$ to $G_s$
9:    **else**
10:       $s_1, s_2 \leftarrow$ two distinct values of $s$ such that $|G_s| < 2\kappa$
11:       Add $i$ to $G_{s_1}$
12:       Add $i$ to $G_{s_2}$
13:    **end if**
14: **end while**
15: **Return** $\mathcal{G} = \{G_s\}_{s \in S}$

---

Note that the counter $i$ never exceeds $\kappa\tau$. To see this, observe that the max value of $i$ is equal to

$$\kappa + \left\lceil \frac{\sum_{s \in S} (2\kappa - \{i : s \in C_i\}|)}{2} \right\rceil = \kappa + \left\lceil \tau\kappa - \frac{1}{2} \sum_{s \in S} |\{i : s \in C_i\}| \right\rceil$$

$$= \tau\kappa + \kappa - \left\lceil \frac{1}{2} \sum_{s \in S} |\{i : s \in C_i\}| \right\rceil$$

$$= \tau\kappa + \kappa - \lceil 1.5\kappa \rceil$$

$$\leq \tau\kappa.$$

Therefore, we choose $N = \lceil \kappa\tau \rceil$ so that $G_s \subseteq N$ as required.

By construction, we also have that $|G_s| = 2\kappa$ for all $G_s \in \mathcal{G}$, so for all $G_s \in \mathcal{G}$,

$$\frac{|G_s|}{|N|} = \frac{2\kappa}{\kappa\tau} = \frac{2}{\tau}.$$

We now show that the Exact Cover by 3-Sets instance has a solution if and only if there exists a $K$ with $|K| \leq \tau/3$ which satisfies $\epsilon$-general proportionality for our instance. Suppose the Exact Cover by 3-Sets has a solution $\tilde{\mathcal{C}} \in \mathcal{C}$. Then $|\tilde{\mathcal{C}}| = \tau/3$. Define the set $K := \{i : C_i \in \tilde{\mathcal{C}}\}$. This $K$ must cover every $s \in S$ exactly once, which by construction of $G_s$ means this $K$ covers every $G_s \in \mathcal{G}$

exactly once. Therefore, we have that $\frac{|K \cap G_s|}{|K|} = \frac{1}{|K|} = \frac{3}{\tau}$ for all $s$. This satisfies

$$\left| \frac{|K \cap G_s|}{|K|} - \frac{|G_s|}{|N|} \right| = \frac{1}{\tau} = \epsilon,$$

so $K$ satisfies $1/\tau$-general proportionality.

To show the other direction, suppose we have a set $K$ with size $|K| \leq \tau/3$ that satisfies $1/\tau$-general proportionality. Because we chose $\epsilon = \frac{1}{\tau}$ and every $G_s$ satisfies $|G_s|/N = 2/\tau$, we must have that $|G_s \cap K| \geq 1$ for all $s$. However, we also know that by construction, any $i \in N$ covers at most three groups $G_s$. This implies that we must have $|K| \geq \tau/3$. Since by assumption $|K| \leq \tau/3$, we therefore must have that $|K| = \tau/3$. Furthermore, the only elements of $N$ that cover 3 groups in $\mathcal{G}$ are the elements $i \in [\kappa]$. Therefore, if $|K| = \tau/3$ and $K$ includes at least one of every group in $\mathcal{G}$, we must have that $K \subseteq [\kappa]$. Finally, combining the facts that every element of $K$ covers exactly three groups, $|K| = \tau/3$, the number of groups is $\tau$, and every group is covered at least once by $K$, we must have that every group is covered exactly once by $K$.

Therefore, we can conclude that $\tilde{\mathcal{C}} = \{C_i : i \in K\}$ must be a solution to the Exact Cover by 3-Sets problem as desired. $\qquad\square$

## L   More Details on Empirical Case Studies

Table 3 summarizes the parameters for each case study.

| Case study | $n$ | $m$ | $k$ for existing subset |
|---|---|---|---|
| BIG-bench | 141 | 120 | 24 |
| HELM | 34 | 67 | 7 |
| Cal Hospital Compare | 50 | 282 | 12 |

Table 3: Summary of case study parameters.

### L.1   Case Study Descriptions and Datasets

We describe each case study in more detail below.

**Case Study 1: BIG-bench [Srivastava et al., 2022].**   We consider the problem of selecting tasks to include in a "Lite" version of BIG-bench . The BIG-bench repository already includes such a subset of tasks called BIG-bench LITE, which includes 24 tasks. BIG-bench in general includes both JSON tasks and programmatic tasks, where JSON tasks are more lightweight to evaluate. Their existing LITE benchmark includes only JSON tasks for ease of evaluation. Thus, for the purposes of this case study, we consider the problem of selecting a subset of the JSON tasks to go in a LITE benchmark. Intuitively, one might want to select the LITE tasks to be representative in some sense of the rest of the JSON tasks, since all tasks are included in the published leaderboards.

*Alternatives:* The BIG-bench repository includes evaluations on its JSON tasks for LLMs of different sizes from three model families: "Big-G", "Big-G sparse", and "GPT". For each individual model, they also include 0-shot, 1-shot, 2-shot, and 3-shot evaluations. For simplicity, we treat each model and each shot count as a separate alternative to be ranked. Thus, we end up with $m = 120$ alternatives.

*Full set:* The full set of tasks we consider consists of $n = 141$ JSON tasks, after filtering to include only the tasks which were evaluated for all alternatives. We consider only JSON tasks in the full set as the existing BIG-bench LITE only includes JSON tasks for ease of evaluation. The list of all tasks is included in the code in the Supplementary Materials. The metric used was always the `preferred_score` field per task.

*Existing subset:* We compare to the existing BIG-bench LITE (BBL) set of tasks, which includes $k = 24$ JSON tasks as of February, 2025. The list of these tasks is included in the code in the Supplemenary Materials and also available at `https://github.com/google/BIG-bench/blob/main/bigbench/benchmark_tasks/keywords_to_tasks.md#big-bench-lite`.

*Dataset:* Data was accessed using the Big-bench API available at `https://github.com/google/BIG-bench`. Code for processing this data is included with the Supplementary Materials. Usage is in compliance with the Apache License, Version 2.0.

**Case Study 2: HELM [Liang et al., 2022].** HELM Classic is another evaluation platform that ranks LLMs based on multiple scenarios and metrics. We consider the problem of selecting a subset of scenarios which is in some sense "representative" of the full set. HELM Lite is an existing variant which includes significantly fewer scenarios than HELM Classic. Note that it is not directly stated that HELM Lite is meant to be representative of HELM Classic (which includes both Core scenarios and other evaluations), but we make this assumption for the purposes of this illustration.

*Alternatives:* We consider $m = 67$ models that appeared on the HELM Classic leaderboard as of February, 2025.

*Full set:* The full set of tasks we consider consists of the accuracy metrics for $n = 34$ scenarios, for which data was posted on the HELM Classic leaderboard. This includes both "Core scenarios" and "Targeted evaluations." A full list of these scenarios is included with the code in the Supplementary Materials.

*Existing subset:* Not all tasks from HELM Lite were included in the HELM Classic leaderboard, and not all models on the HELM Classic leaderboard were evaluated on the HELM Lite leaderboard. Thus, as our existing subset baseline, we selected the subset of HELM Classic scenarios which were also included as HELM Lite scenarios.

This yielded a set of $k = 7$ scenarios, which included: *NarrativeQA, NaturalQuestions (open-book), NaturalQuestions (closed-book), OpenbookQA, MMLU (Massive Multitask Language Understanding), MATH*, and *GSM8K (Grade School Math)*.

*Dataset:* Data was downloaded as HTML tables in February, 2025 from the following links:

- `https://crfm.stanford.edu/helm/classic/latest/#/leaderboard/core_scenarios`

- `//crfm.stanford.edu/helm/classic/latest/#/leaderboard/targeted_evaluations`

- `https://crfm.stanford.edu/helm/lite/latest/#/leaderboard/core_scenarios`

Leaderboard data can also be accessed at `https://github.com/stanford-crfm/helm`. Only accuracy metrics were considered. Code for processing this data is included with the Supplemental Materials. Usage is in compliance with the Apache License, Version 2.0.

**Case Study 3: Cal Hospital Compare [Cal Hospital Compare, 2025].** Finally, we demonstrate how our methods can apply in evaluation settings beyond LLMs by considering the problem of hospital quality evaluation. Cal Hospital Compare is a platform that awards a "patient safety honor roll" status to hospitals in California that perform particularly well in a set of quality measures collected from the Centers for Medicare and Medicaid Services (CMS). The honor roll selection procedure considers 12 hospital quality measures drawn from the CMS Hospital Compare database, which collects hundreds of measures from hospitals throughout the US. A continuing challenge is selecting these 12 quality measures, which Cal Hospital Compare identifies as a "an 18-month, multistakeholder process of rigorously evaluating existing national patient safety measures."[4] For the purposes of this illustration, we consider the problem of selecting a set of quality measures which is "representative" of existing patient safety measures available in the CMS Hospital Compare database. According to the "algorithmic approach," Cal Hospital Compare awards honor roll status to eligible hospitals "with two-thirds of their measures above the 50th percentile of good performance (and none below the 10th percentile)."

*Alternatives:* We consider $m = 282$ hospitals in California which were eligible for algorithmic evaluation according the criteria specified by Cal Hospital Compare. Specifically, they had scores for at least 6 of the currently selected 12 measures.

*Full set:* We consider a full set of $n = 50$ patient safety measures available through CMS Hospital Compare. These were selected to include only the measures that were directly related to the categories from which the original 12 measures were selected, so as not to include measures unrelated to patient safety. A full list of these measures is included in the code submission. It is possible that some

---

[4] `https://calhospitalcompare.org/wp-content/uploads/2025/04/FactSheet_Patient-Safety-Honor-Roll-List_Cal-Hospital-Compare_2025-1.pdf`

important measures were missed in this illustration, and any real practical application should bring in additional domain expertise to carefully tailor the full set to precise practical goals.

*Existing subset:* We compare against the existing $k = 12$ measures currently used by Cal Hospital Compare for their patient safety honor roll. A full list of these measures is shown in Figure 3.

| Domain | Source | Measurement Period |
|---|---|---|
| Healthcare Associated Infections 
• CLABSI 
• CAUTI 
• SSI Colon Surgery 
• MRSA 
• CDI | CMS Hospital Compare (based on unnormalized HAI data) | 1/1/2023 - 12/31/2023 |
| AHRQ PSI 90 | CMS Hospital Compare | 7/1/2021 – 6/30/2023 |
| Sepsis Management | CMS Hospital Compare | 1/1/2023 - 12/31/2023 |
| Patient Experience HCAHPS Score 
• Nurses always communicated well 
• Doctors always communicated well 
• Always received help as soon as wanted 
• Staff always explained about medicines 
• Patients understood their care when they left the hospital | CMS Hospital Compare | 1/1/2023 - 12/31/2023 |
| Hospital Letter Grade | Leapfrog Hospital Safety Grades | Average GPA from the last three reporting periods (Fall 2023, Spring 2024, Fall 2024) |

Figure 3: Table of Patient Safety Honor Roll Measures published by Cal Hospital Compare (https://calhospitalcompare.org/wp-content/uploads/2025/04/FactSheet_Patient-Safety-Honor-Roll-List_Cal-Hospital-Compare_2025-1.pdf). As our "existing subset", we consider the set of $k = 12$ measures selected from CMS Hospital Compare.

*Dataset:* Data was downloaded as CSV files in February, 2025 from the CMS Hospital Compare provider data repository (https://data.cms.gov/provider-data/). A full directory of all available datasets can be found at https://data.cms.gov/provider-data/dataset/dgmq-aat3#data-dictionary. The specific datasets used were:

- `Healthcare_Associated_Infections-Hospital.csv`
- `Complications_and_Deaths-Hospital.csv`
- `Timely_and_Effective_Care-Hospital.csv`
- `HCAHPS-Hospital.csv`
- `Maternal_Health-Hospital.csv`
- `Unplanned_Hospital_Visits-Hospital.csv`

This data is part of the public domain. Code to aggregate and process these datasets is included with the Supplemental Materials.

### L.2 Integer program computation

Integer programs were solved using CPLEX. A maximum solve time of 10 minutes was set for each integer program. Thus, it is possible that the integer programming solutions were suboptimal when this limit was reached. This limit was only reached on BIG-bench.

All experiments were run on a MacBook Pro with an Intel Core i7.

