# OpenReview forum: "Metritocracy: Representative Metrics for Lite Benchmarks"
_NeurIPS.cc/2025/Conference — NeurIPS 2025 poster_

### Official Review · Reviewer_KYHN · 2025-07-02

**Clarity:** 2
**Significance:** 2
**Originality:** 2
**Rating:** 4
**Confidence:** 4

**Summary:**

This paper addresses the problem of selecting a "representative" subset of metrics from a full suite for efficient evaluation, particularly in contexts like LLM assessment and hospital quality evaluation, where using all metrics is computationally expensive or hard to interpret. Key contributions include:
1. Formalizing two notions of representation: Positional representation ensures every alternative is sufficiently represented at each rank cutoff, parameterized by a group size $ g $; positional proportionality guarantees no alternative is over- or under-represented by more than an additive error $ \epsilon $ at any position.
2. Theoretical bounds: For positional representation, tight (up to logarithmic factors) upper and lower bounds on the minimum number of metrics needed are established, with a polynomial-time greedy algorithm achieving the upper bound. For positional proportionality, tight (up to constants) bounds are provided, and it is shown to approximate social choice scoring rules.
3. Generalizations: Extending both properties to handle pre-defined groups of metrics, with proofs that finding the smallest satisfying subset is NP-hard, while bounds from the base cases extend to these generalizations.
4. Empirical validation: Case studies on BIG-bench, HELM, and hospital quality evaluation demonstrate that the proposed methods outperform existing "lite" subsets in efficiency while maintaining representativeness, and can augment existing subsets to improve performance.

**Questions:**

1. The paper highlights the lack of a formal definition for "representativeness" in metric subset selection, while existing lite benchmarks (e.g., BIG-bench Lite, HELM Lite) rely on manual curation. What specific flaws in representativeness do these existing subsets exhibit? For example, are there systematic under-representations of certain alternatives? Does the paper quantify these flaws using empirical data?
2. The paper draws parallels between LLM evaluation and hospital quality assessment as scenarios requiring representative metric subsets. However, LLM metrics and hospital metrics differ significantly in semantics, distribution, and evaluation goals. How does the paper justify generalizing the proposed framework across such diverse domains? Are adjustments to parameters (e.g., $g$ or $\epsilon$) needed for domain-specific contexts?
3. Positional Representation requires that if an alternative $a$ is ranked in the top $r^2$ positions by $\ell \cdot g$ metrics in $N$, it must be ranked in the top $r$ positions by at least $\ell$ metrics in $K$. Why is $r^2$ chosen as the positional mapping? Is this choice theoretically validated or empirically tested? How would switching to a linear relationship $r$ affect the theoretical bounds (upper and lower)?
4. The paper claims that Positional Proportionality approximates any social choice scoring rule with an error bounded by $\epsilon$. However, scoring rules weight ranks differently, Borda assigns higher weights to top positions. How does Positional Proportionality adapt to such weight variations? Are there scoring rules that cannot be effectively approximated?
5. The greedy algorithm for Positional Representation Alg 1 uses a "coloring" step where every $g$ uncolored instances of an alternative are labeled with the same color. What is the rationale for this coloring strategy? Could alternative strategies, e.g., grouping by rank positions, yield better subsets?
6. Positional Proportionality uses an additive error $\epsilon$, with bounds scaling with $1/\epsilon^2 \log(m)$. Why is additive error preferred over multiplicative error? In domains like hospital quality assessment, relative errors in metric proportions may be more critical—does the framework need extension to support multiplicative error?
7. The paper notes that Positional Representation and Positional Proportionality are "mutually non-implied," with examples showing Positional Proportionality can be more efficient, e.g., $|K|=1$ when all metrics agree. However, in scenarios with highly conflicting rankings, which property better ensures representativeness? Can experimental data compare their applicability?
8. Figure 1 shows "minimum $g$ for existing subsets", but the calculation method is unclear. How is the smallest $g$ for which an existing subset satisfies Positional Representation determined? Does this account for biases in manual curation?
9. In the hospital quality case study, adding <5 metrics to the existing 12 significantly reduces $\epsilon$. What criteria guided the selection of these additional metrics? Was domain knowledge incorporated? Would results remain robust if selection were fully algorithmic?

**Ethical Concerns:**

["NO or VERY MINOR ethics concerns only"]

**Final Justification:**

Thank you for addressing my concerns, and I will raise my score accordingly.

**Limitations:**

Refer to Questions

**Quality:**

2

**Strengths And Weaknesses:**

Refer to Questions

---

> ### Author Rebuttal · Authors · 2025-07-28
>
> We thank the reviewer for the detailed list of questions. Some of these questions seem to stem from misunderstandings or are already addressed in the paper, as we explain below.
>
> > *1. The paper highlights the lack of a formal definition for "representativeness" in metric subset selection, while existing lite benchmarks [...] rely on manual curation. What specific flaws in representativeness do these existing subsets exhibit? For example, are there systematic under-representations of certain alternatives? Does the paper quantify these flaws using empirical data?*
>
> Given the lack of existing formalism for representation of metrics, our paper serves three purposes: with our new formalism, (1) there now exists a quantitative method to check to what degree existing manually curated metrics are representative, and (2) if not representative enough, an algorithmic intervention to add new metrics to improve representation. Additionally, in domains where existing manually curated “lite” metric subsets don’t exist, (3) our algorithm can propose a subset of metrics from scratch that can serve as a base to seed additional manual curation from domain experts, thus saving human hours.
>
> Regarding existing manually curated metrics: we indeed demonstrate empirically that existing manually curated metrics may not be representative by the definitions we provided, depending on the tolerance for error. The dashed lines in Figures 1 and 2 show the minimum group size $g$ and tolerance $\epsilon$ for which the manually curated subsets satisfy the representation definition. For any $g$ and $\epsilon$ lower than that, the manually curated sets would not be representative.
>
> We also show that it’s possible to significantly improve representation by adding relatively few additional metrics using our proposed algorithms: for example, Figure 2 shows that on the HELM dataset, $\epsilon$ can be halved by adding four metrics to the existing manually curated lite set.
>
> > *2. The paper draws parallels between LLM evaluation and hospital quality assessment [...]. However, LLM metrics and hospital metrics differ significantly in semantics, distribution, and evaluation goals. How does the paper justify generalizing the proposed framework across such diverse domains? Are adjustments to parameters (e.g.,  or ) needed for domain-specific contexts?*
>
> The core evaluation goals are actually quite similar between public LLM evaluation platforms and the hospital honor roll: the evaluator wants to reward and encourage good performance, and the evaluator wants the evaluations to be useful in some sense to stakeholders, like users or patients. Of course, how “strict” a notion of representation an evaluator needs absolutely depends on the specific application, and the parameters $g$ or $\epsilon$ must be chosen by the evaluator to reflect this. It’s possible that stricter representation is more important in the healthcare domain: if $\epsilon$ is not low enough, a high-performing hospital might not make it onto the honor roll, and a hospital might get on the honor roll that actually doesn’t perform well. Whether one would use PR or PP also depends on the domain: with LLMs leaderboards, PR might be enough to ensure that LLMs that are good across a variety of tasks are ranked highly in the lite benchmark set. On the other hand, PP might be important in healthcare to ensure that one can identify hospitals that perform poorly using the lite metric set. To summarize, our framework and results provide sufficient flexibility to apply to diverse domains.
>
> > *3. Positional Representation requires that if an alternative is ranked in the top $r^2$ positions by $\ell \cdot g$ metrics in N, it must be ranked in the top positions by at least  metrics $\ell$ in $K$. Why is  chosen as the positional mapping? Is this choice theoretically validated or empirically tested? How would switching to a linear relationship affect the theoretical bounds (upper and lower)?*
>
> Positional Representation as defined in Definition 1 says that if an alternative is ranked in the top $r$ positions by $\ell \cdot g$ metrics in $N$, then it must be ranked in the top $r$ positions by at least $\ell$ metrics in $K$. This is already a linear relationship (and we don't believe we use $r^2$ anywhere in the paper). A linear relationship is conceptually natural because it guarantees *proportional* representation for every position. Finally, we prove (Theorems 2 and 3) that the linear relationship is (almost) tight, so in this sense the choice is indeed "theoretically validated."
>
> > *4. The paper claims that Positional Proportionality approximates any social choice scoring rule with an error bounded by $\epsilon$. However, scoring rules weight ranks differently, Borda assigns higher weights to top positions. How does Positional Proportionality adapt to such weight variations? Are there scoring rules that cannot be effectively approximated?*
>
> Theorem 7 states that positional proportionality guarantees an additive approximation of the scores of every normalized scoring rule. Informally, this means that for any alternative, that alternative's *score* when evaluating the scoring rule on $N$ is within $\epsilon$ of that alternative's *score* when evaluating the scoring rule on just $K$. Therefore, every normalized scoring rule (including Borda) can be approximated, no matter how the scoring rule  weights the ranks.
>
>
> > *5. The greedy algorithm for Positional Representation Alg 1 uses a "coloring" step[...]. What is the rationale for this coloring strategy? Could alternative strategies, e.g., grouping by rank positions, yield better subsets?*
>
> The greedy algorithm is guaranteed to return a subset $K$ that satisfies positional representation as shown in Theorem 3, and by Theorem 2 this is within a logarithmic factor of being optimal. Therefore, no other algorithm can in the worst-case yield better subsets than Algorithm 1. The coloring approach is motivated by the greedy algorithm for set cover in theoretical computer science, which solves a related problem and gives strong approximation guarantees.
>
> > *6. Positional Proportionality uses an additive error $\epsilon$,[...]. Why is additive error preferred over multiplicative error? In domains like hospital quality assessment, relative errors in metric proportions may be more critical—does the framework need extension to support multiplicative error?*
>
> If we were to require a multiplicative factor of approximation, then there exist instances of the problem where no subsets of $N$ except for $N$ itself actually satisfy the property for any $\epsilon \le 1$. Formally, suppose we want that $|\frac{C(N,r,a)}{|N|} - \frac{C(K,r,a)}{|N|}| \le \epsilon \frac{C(N,r,a)}{|N|}$. Then suppose we have that $n = m$ and that every alternative is ranked first by exactly one benchmark. In order to satisfy the multiplicative property described above, we must have that each alternative is ranked first by at least one benchmark in $K$, which requires that $K = N$. This example shows that a multiplicative form of PP is too strong of a requirement.
>
> > *7. The paper notes that Positional Representation and Positional Proportionality are "mutually non-implied,"[...]. However, in scenarios with highly conflicting rankings, which property better ensures representativeness? Can experimental data compare their applicability?*
>
> For this problem, there is no notion of "better" representativeness. Positional representation and positional proportionality are two *different types* of representativeness, each with different interpretations and theoretical guarantees described in the paper. In Figures 1 and 2, we have experimental results on the same data sets for both notions of fairness, but in the end it is up to a user to decide which is the form of representativeness they desire.
>
> > *8. Figure 1 shows "minimum  for existing subsets", but the calculation method is unclear. How is the smallest  for which an existing subset satisfies Positional Representation determined? Does this account for biases in manual curation?*
>
> For a given existing subset of metrics $K$ (e.g. $K =$ the existing BIG-bench Lite metrics), the “minimum $g$ for the existing subset” plotted in Figure 1 is the minimum group size $g$ such that Definition 1 is satisfied. That is, we plot $\min\{g : \forall r \in [1:m], \forall a \in A, C(K,r,a) \ge \left \lfloor \frac{C(N,r,a)}{g} \right \rfloor\}$. Code for calculating this is included with the code submission, and is simply done by binary search over $g$. Any biases in manual curation will be reflected in the resulting minimum value of $g$: a more “biased” manual curation (relative to the existing PR definition) will end up with a higher minimum value of $g$.
>
> > *9. What criteria guided the selection of [the additional metrics in the hospital quality case study]? Was domain knowledge incorporated? Would results remain robust if selection were fully algorithmic?*
>
> The additional metrics were selected by solving the following optimization problem: What is the minimal set of additional metrics that can be added to the existing 12 metrics such that PP or PR is satisfied for each given value of $g$ or $\epsilon$? This is solved using the same algorithms proposed (Algorithm 1 and integer programming).
>
> The algorithmic selection of these additional metrics can be seen as a hybrid approach that uses our algorithms to augment domain knowledge rather than replace existing human metric selection processes. We emphasize that we *do not* argue that human curation should be replaced, but rather that our algorithm and formalism can be an effective tool to *support* domain experts. A comparison to fully algorithmic selection is also included in Figures 1 and 2, which we show is able to achieve stricter representation using fewer metrics. This fully algorithmic selection can be used to create a starting point for a Lite metric set from which human domain experts can build.

---

> > ### Comment · Reviewer_KYHN · 2025-08-05
> >
> > Thank you for addressing my concerns, and I will raise my score accordingly.

---

### Official Review · Reviewer_GiFD · 2025-07-02

**Clarity:** 4
**Significance:** 3
**Originality:** 3
**Rating:** 5
**Confidence:** 3

**Summary:**

This paper introduces and studies the problem of finding a small representative set of metrics $K$ from a larger set of metrics $N$ specified by the user. The goal is for $K$ to be a ``good approximation’’ of $N$. Drawing on ideas from computational social choice, the authors formalize two notions of “representativeness” for $K\subseteq N$. The first notion – positional representation – requires that for every rank cutoff $r$ and alternative $a$, if at least $g\ell$ of the $n$ metrics rank $a$ in the top $r$, then at least $\ell$ metrics in $K$ must do so too. The second notion, positional proportionality, extends this to a two-sided requirement. The authors also extend both notions to. The authors also extend both notions to The setting where one has to be representative of different subsets of metrics $G_1, G_2,\dots\subseteqq N$.

For the first notion, the paper shows that a lower bound on the size of the selected set K and an efficient algorithm that matches this size up to a logarithmic factor. For the second notion, the paper provides lower and upper bounds on the size of the select set of metrics required, and matches it where the lower and upper bounds match up to constant factors.

There seems to be an interesting fundamental problem in social choice theory that has not received much attention, to the best of my knowledge. The authors claim that this also has applications in evaluating LLMs.

**Questions:**

In the paper, different notions of positional representation typically look at item-wise guarantees. Is it possible to extend the techniques in this work when we require, for instance, pairs of items to be treated similarly by the sub-small subset of metrics selected and the large input subset of metrics?

Since the notions of representation studied in this work assume that the underlying metrics output a ranking, I think they cannot directly be applied to metrics which output scores for individuals, such as Borda. Can the authors comment on extensions to such score-based voting rules?

**Ethical Concerns:**

["NO or VERY MINOR ethics concerns only"]

**Final Justification:**

I keep my initial score as I think the paper studies an interesting question. I still believe that it would be nicer to present the fundamental question directly without delving too much into the connection to LLMs, which seems weak. I thank the authors for answering the questions I raised.

**Limitations:**

Yes

**Quality:**

3

**Strengths And Weaknesses:**

Strengths:

1. The paper introduces an interesting problem in social choice theory and provides clean theoretical results for it.
2. The paper is well-written and easy to follow.
3. The paper evaluates its proposed solutions using carefully designed empirical studies.

Weaknesses:

1. While the authors chiefly motivate their problem through its applications in LLM evaluation, I think its application is weak. And instead of focusing on the fundamental nature of the problem might make the paper more grounded and interesting.

---

> ### Author Rebuttal · Authors · 2025-07-28
>
> > *In the paper, different notions of positional representation typically look at item-wise guarantees. Is it possible to extend the techniques in this work when we require, for instance, pairs of items to be treated similarly by the sub-small subset of metrics selected and the large input subset of metrics?*
>
> This is an interesting question, and the answer depends on what you mean by "treated similarly" for a pair of items. For example, suppose we have alternatives $a$ and $b$ that are 'clones' of each other so they are ranked adjacent to each other in every benchmark. Then positional proporionality immediately guarantees that the fraction of time $a$ and $b$ are ranked at position $r$ in the subset $K$ will be within $2\epsilon$ for every $r$. The same will approximately hold even if $a$ and $b$ aren't perfect clones of each other and are only approximate clones of each other. Now if we want to guarantee that $a$ and $b$ have a similar representation in $K$ when $a$ and $b$ have very different rankings in $N$, then that becomes much more difficult. In fact, $a$ and $b$ having similar representation in $K$ while having different representation in $N$ would directly contradict our notions of position proportionality and position representation. Therefore, such a guarantee would not be compatible with our notions of fair representation.
>
> > *Since the notions of representation studied in this work assume that the underlying metrics output a ranking, I think they cannot directly be applied to metrics which output scores for individuals, such as Borda. Can the authors comment on extensions to such score-based voting rules?*
>
> Any metric that outputs scores for individuals can be directly converted to a ranking (by score), in which case the notions of representation studied in this work can be directly applied. That being said, doing this effectively "throws away" any information in the scores for a metric. Another option is, if a metric involves a Borda count generated from a bunch of submetrics, then we can instead guarantee positional representation/proportionality with respect to the entire set of submetrics.
>
> We do also have results in our paper for score-based aggregation rules of benchmarks; see Theorem 7, which informally says that if $K$ satisfies position proportionality, then any scoring rule (e.g. Borda) evaluated on $K$ will be a good approximation for the same score-based rule evaluated on $N$.

---

> > ### Comment · Reviewer_GiFD · 2025-08-05
> >
> > Thank you for taking the time to answer my questions. While the connection to LLMs seems a bit superficial, I still think the problem studied here is interesting;

---

### Official Review · Reviewer_8Kid · 2025-07-03

**Clarity:** 3
**Significance:** 4
**Originality:** 4
**Rating:** 5
**Confidence:** 3

**Summary:**

This paper tackles the problem of selecting a representative subset of evaluation metrics from a larger set, particularly in settings like LLM or hospital quality evaluations. The authors formalize two new concepts -- positional representation and positional proportionality-- inspired by social choice theory. They derive tight upper/lower bounds, prove NP-hardness, and propose practical algorithms for subset selection. Their approach is validated via three real-world case studies: BIG-bench, HELM, and Cal Hospital Compare.

**Questions:**

1. Incorporating metric importance or cost.

How would the framework extend to settings where metrics have differing levels of importance, quality, or computational cost?

2. Integrating other constraints.

Can fairness, diversity, or cost-sensitive constraints be integrated into the subset selection procedure (e.g., ensuring representation across language groups or domains)?

**Ethical Concerns:**

["NO or VERY MINOR ethics concerns only"]

**Limitations:**

yes

**Paper Formatting Concerns:**

no concerns

**Quality:**

4

**Strengths And Weaknesses:**

## Strengths
1. Quality: Strong theoretical analysis with tight bounds and clear proofs; algorithms are sound and practically applicable.
2. Originality: Introduces novel formalizations of metric representativeness based on social choice theory—fresh perspective in benchmarking.
3. Significance: Addresses a real need in scalable, interpretable benchmark design; applicable to various domains.
4. Clarity: Generally well-written and organized, with intuitive examples and thoughtful discussion.

## Weaknesses
1. Bias inheritance from full metric set: The method assumes the full set of metrics is well-balanced. If the original set is biased or overrepresents certain aspects, the selected subset will likely reflect those biases as well.
2. Limited robustness to distribution shift: While the subset can be efficiently recomputed, it may still fail to generalize well when evaluating substantially different or unseen alternatives. In such cases, user intervention (e.g., choosing new values for g or \epsilon) may be needed repeatedly.
3. Scalability of IP: Integer programming guarantees optimal subsets but may become infeasible in large-scale settings with hundreds or thousands of metrics.
4. Assumption of complete rankings: The framework assumes that all metrics provide complete rankings over alternatives, which is not always true in real-world LLM evaluations where missing data is common.

---

> ### Author Rebuttal · Authors · 2025-07-28
>
> > *How would the framework extend to settings where metrics have differing levels of importance, quality, or computational cost?*
>
> This is a great question, and our results can in some cases incorporate these additional aspects. Suppose that instead of caring about the size of the subset of metrics, we instead have a computational cost for each metric, and our goal is to keep the total computational cost below some threshold. In this case, a user could easily modify the integer programs in Appendix A to include weights in the objective function. For a more computationally efficient solution, you could also run the greedy algorithm and select the set that gives the lowest cost per new color (this mimics the greedy algorithm for the weighted set cover problem). This algorithm would run in polynomial time, and give a $\log(mn/g)$ approximation to the optimal. However, depending on the costs of the benchmarks, there no longer exists a nice guarantee for the total cost of $K$ as in Theorem 3. We are happy to include further discussion of this in the final version of the paper.
>
> If there is an importance weighting of benchmarks then we functionally have two objectives: minimizing the size of $K$ and maximizing the sum of importance weights. Our results do not directly generalize to this multi-objective setting — this is an interesting question for future work.
>
> > *Can fairness, diversity, or cost-sensitive constraints be integrated into the subset selection procedure (e.g., ensuring representation across language groups or domains)?*
>
> Yes! Using the generalized properties defined in Section 4, a user can use our subset selection procedure for any general form of representation. As in your example, we could choose the collection of groups $\mathcal{G}$ to include groups of different languages, in which case any set $K$ satisfying generalized representation/proportionality will ensure repesentation for every language. Similarly, if we choose $\mathcal{G}$ to include sets of benchmarks from different domains, then these properties guarantee representation for these groups. See Lines 293-298 for some explicit examples of how we can use these general properties. All of the results from Sections 2 and 3 generalize to this broader setting, which we show rigorously in Appendix B.

---

> > ### Comment · Reviewer_8Kid · 2025-08-05
> >
> > Thank you for addressing my questions. My concerns have been resolved, and I will maintain my current score.

---

### Comment · Area_Chair_nujq · 2025-08-04

Dear Reviewers,

As we near the end of the rebuttal period, this is a friendly reminder to submit your responses to the authors by **August 6**. Your engagement is crucial for our final decision.

**Action Required**

- Read the rebuttal carefully: Authors have invested significant effort in addressing your concerns.

- Reply directly to authors: Briefly acknowledge their points and indicate whether your assessment has changed (or why it remains unchanged).

- Update your review (if applicable): Adjust scores/comments in the review system to reflect your current stance.

Your AC

---

### Decision · Program_Chairs · 2025-09-17

**Decision:**

Accept (poster)

**Comment:**

This paper focuses on the problem of selecting representative subsets of evaluation metrics from larger suites, with applications in LLM evaluation and hospital quality assessment. It introduces two novel notions—positional representation and positional proportionality—grounded in social choice theory, and provides tight theoretical bounds, efficient algorithms, and validation through real-world case studies on BIG-bench, HELM, and Cal Hospital Compare.

The paper is generally well-executed and offers several notable strengths.

-  It provides strong theoretical analysis with tight bounds and rigorous proofs.
-  It introduces novel definitions of representativeness inspired by social choice theory, bringing a fresh perspective to benchmark design.
-  The proposed methods address a practical need for scalable, interpretable evaluation across domains.

After the rebuttal, most of the reviewers' concerns were adequately resolved. The authors clarified how the framework can incorporate importance weights, costs, fairness constraints, and generalization across domains, while also addressing reviewer questions about representativeness in existing curated subsets. Reviewers confirmed that their concerns were largely addressed.

Overall, I recommend accept. The paper is technically solid and makes both theoretical and applied contributions. For the final version, the authors should incorporate reviewer suggestions, especially clarifying domain-specific applicability and robustness to potential biases in the full metric set.